# REFEREVERYTHING: TOWARDS SEGMENTING EVERYTHING WE CAN SPEAK OF IN VIDEOS

## ABSTRACT

We present REM (Refer Everything Model), a framework for segmenting a wide range of concepts in video that can be described through natural language. To achieve this level of generalization, our method capitalizes on visual-language representations learned by video diffusion models on Internet-scale datasets. A key insight of our approach is preserving as much of the generative model's original representation as possible, while fine-tuning it on narrow-domain Referring Object Segmentation datasets. As a result, despite being exclusively trained on object masks from a limited set of categories, our framework is able to accurately segment and track both rare, unseen objects and non-object, dynamic concepts, such as waves crashing in the ocean. To better quantify the generalization capabilities of our model, we introduce a new benchmark for Referring Video Process Segmentation (RVPS), which captures dynamic phenomena that exist at the intersection of video and language. Our experiments show that REM performs comparably to state-of-the-art approaches on in-domain datasets while outperforming them by up to 28% out-of-domain, leveraging the power of Internet-scale pretraining. We include all of the video visualizations at this anonymous page.

## 1 INTRODUCTION

One of the most remarkable features of natural language is its ability to describe human visual experience in all of its richness and complexity. Whether capturing fleeting moments, like raindrops rolling down the window, or smoke dissipating from a cigarette (see row 2 in Figure 1), or describing dynamic processes, such as a glass shattering or a whirlpool forming in the water (row 1 in Figure 1), if we can utter them, we can also accurately localize them in space and time. This universal mapping between the discrete, symbolic realm of language and the continuous, ever-changing visual world is developed through a lifetime of visual-linguistic interaction (Barsalou, 1999; Popham et al., 2021).

The corresponding problem in computer vision - Referring Video Segmentation (RVS) (Gavrilyuk et al., 2018; Hu et al., 2016), is defined as the task of segmenting a specific region in a video based on a natural language description. However, virtually all existing benchmarks and methods focus on a specific subset of RVS - Referring Video Object Segmentation (RVOS) (Seo et al., 2020; Wu et al., 2022a), where the goal is to track and segment the *object* referenced by a given expression. Why has the field concentrated so narrowly on this task? Although multiple factors contribute, we argue that the primary reason lies in the data. Historically, RVOS datasets have been developed by adding referring expression annotations to existing object tracking benchmarks (Pont-Tuset et al., 2017; Xu et al., 2018), which are inherently object-centric and limited in scale.

At the same time, recent advances in Internet-scale datasets with billions of paired image- and video-language samples (Schuhmann et al., 2022; Bain et al., 2021) have opened new possibilities. These datasets have been used to train powerful denoising diffusion models (Rombach et al., 2022; Wang et al., 2023), and provide excellent representations of the natural visual-language manifold. In the image domain, numerous studies have shown that re-purposing diffusion models can yield highly generalizable representations of object shapes (Zhao et al., 2023; Ozguroglu et al., 2024). Very recently, Zhu et al. (2024) explored the application of video diffusion models for referring segmentation, but their approach exhibited limited generalization capabilities.

In this work, we introduce a novel approach to RVS that leverages large-scale video-language representations learned by diffusion models. Our method, described in Section 3 and shown in Figure 3,

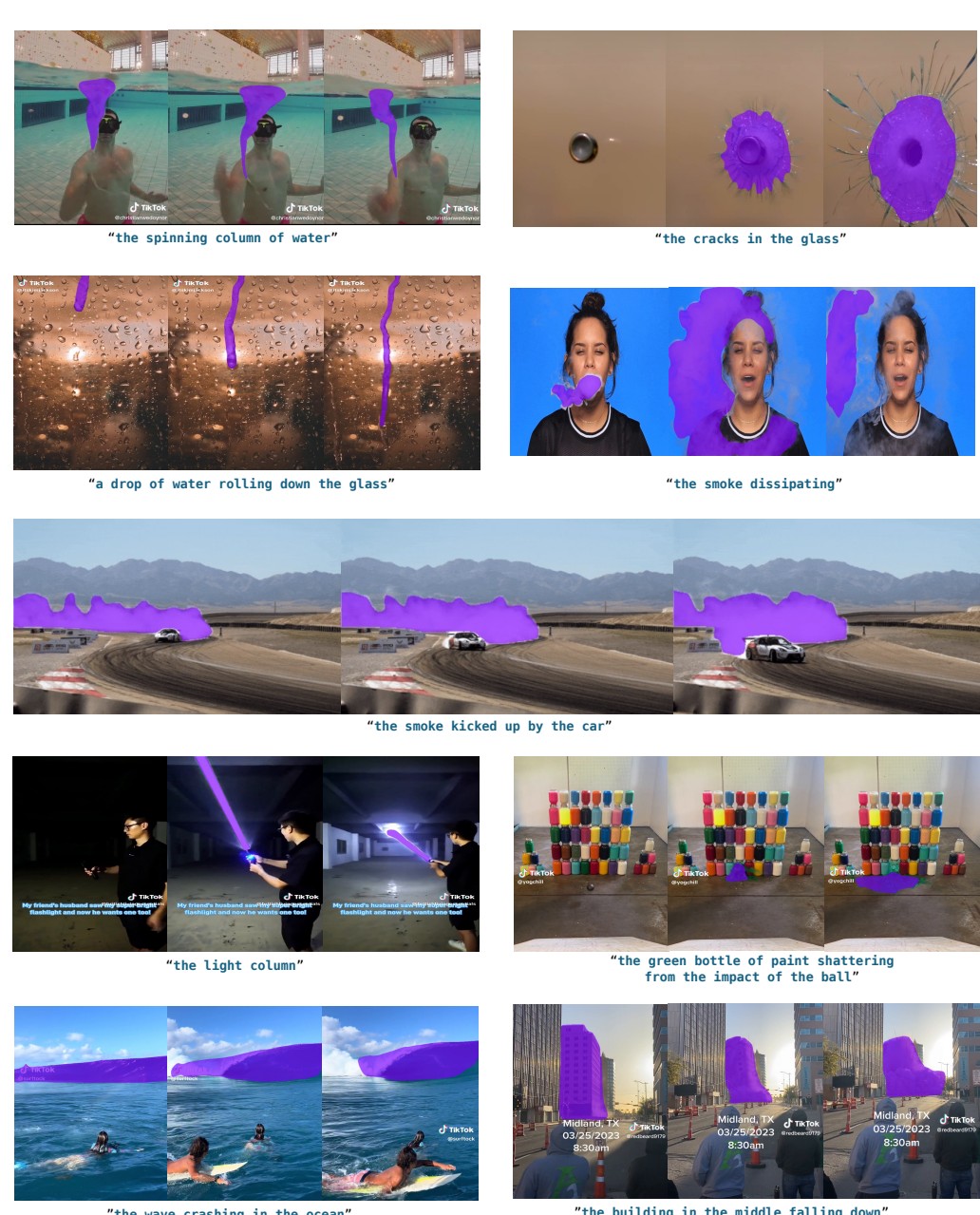

Figure 1: Sample results of our method, which can segment a wide range of concepts in video that can be described through natural language by capitalizing on visual-language representations learned by video diffusion models. REM generalizes with ease to challenging, dynamic concepts, such as raindrops or shattering glass, as shown above. Video visualizations are available here.

enables spatio-temporal localization of a wide range of concepts (Ghorbani et al., 2019) in video that can be described through natural language. A key factor behind our approach's success is preserving the rich representations learned by the generative model (see Figure 2). To achieve this, we retain the original model architecture and fine-tune it on existing referring image- and video-segmentation datasets, adjusting the output to generate object masks instead of Gaussian noise. As shown in Section 5.1, our model demonstrates competitive performance against specialized state-of-the-art models, as well as recent diffusion-based methods, on RVOS benchmarks. More significantly, it exhibits a much stronger generalization to unseen object categories and non-object dynamic concepts.

To quantify this effect, we report results on the open-world object tracking benchmark - BURST (Athar et al., 2023), as well as collect a new benchmark that focuses on dynamic pro-

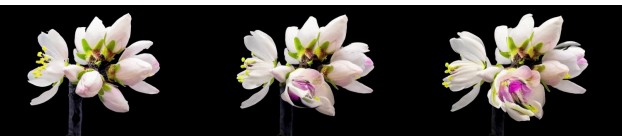

"Explode colorful smoke coming out"          "Time-lapse of a blooming flower on a steam "

Figure 2: Through Internet-scale pre-training, video diffusion models can generate realistic videos capturing the entire diversity of the dynamic visual world (generated samples shown above). We leverage their powerful visual-language representation for open-world referring video segmentation.

cess in Section 4. We define the latter as temporally evolving events, where the subjects undergo continuous changes in state, shape, or appearance (see examples in Figure 1). Many of these concepts are best captured through the combination of video and language, as language helps define them, while temporal information is crucial for accurate localization. Our new benchmark, which we call RVPS for Referring Video Process Segmentation, consists of 111 videos that are labeled with referring expressions and masks at 24 fps and span 38 unique concepts. Our experiments in Section 5.2 demonstrate that traditional RVOS approaches fail to generalize to this challenging scenario, whereas our method effortlessly segments a wide spectrum of concepts, from light reflections to objects dramatically changing appearance (see Figures 1 and 4).

Crucially, our approach strongly outperforms the very recent method of Zhu et al. (2024), which is also based on a video-diffusion representation, by up to 28%. We investigate this in Section 5.3 and experimentally demonstrate that preserving as much of the representation learned during generative pre-training as possible is key for achieving the highest degree of generalization in referring video segmentation. We will release the code, models, and data for reproducing our results.

## 2 RELATED WORK

**Referring Video Segmentation (RVS)** involves segmenting specific regions in a video based on a natural language description (Gavrilyuk et al., 2018; Khoreva et al., 2019; Seo et al., 2020). Most benchmarks for this task were developed by adding referring expression annotations to existing Video Object Segmentation (VOS) datasets, such as DAVIS'17 (Pont-Tuset et al., 2017) or YouTube-VOS (Xu et al., 2018). Consequently, the role of language in these benchmarks is limited to providing an interface for user-initialized object tracking (Wu et al., 2013; Perazzi et al., 2016). While this specific task — Referring Video Object Segmentation (RVOS) — is valuable, it addresses only a narrow subset of the possible interactions between language and the space-time continuum of videos. Equally important is the ability of RVS methods to segment video concepts beyond common object categories. To address this gap, we introduce a new benchmark focused on segmenting dynamic processes, which we term Referring Video Process Segmentation (RVPS).

Earlier RVOS approaches (Bellver et al., 2020; Ning et al., 2020; Hui et al., 2021) generally employed a *bottom-up* strategy: first, image-level methods (Rother et al., 2004; Ye et al., 2019; Carion et al., 2020; Plummer et al., 2015) were applied to obtain frame-level masks, followed by spatio-temporal reasoning, such as mask propagation (Seo et al., 2020), to refine the segmentation across frames. More recently, with the success of cross-attention-based methods (Vaswani, 2017; Meinhardt et al., 2022; Zeng et al., 2022) in object segmentation and tracking, query-based architectures have been introduced to RVOS, leading to significant improvements, with ReferFormer (Wu et al., 2022a) and MUTR (Yan et al., 2024) being notable examples. The limited scale of paired video-language data with segmentation annotations has always been a major limitation in RVOS, causing most methods to train jointly on video and image samples (Kazemzadeh et al., 2014; Jhuang et al., 2013). The latest approaches go even further and unify all object localization datasets and tasks in a single framework to maximize the amount of training data (Yan et al., 2023; Wu et al., 2024; Cheng et al., 2023). However, while these models excel in object tracking, they struggle to generalize to more dynamic concepts. In contrast, we demonstrate that generative video-language pre-training on Internet-scale data (Schuhmann et al., 2022; Bain et al., 2021) results in a universal (i.e. not limited to one domain) mapping between the space of language and the ever-changing visual world.

**Diffusion Models** have become the de-facto standard for generative learning in computer vision (Sohl-Dickstein et al., 2015; Ho et al., 2020) and beyond (Chi et al., 2023). Among them, the Denoising Diffusion Probabilistic Model (DDPM) (Ho et al., 2020) leverages neural network

components to model the denoising process and builds a weighted variational bound for optimization. Stable Diffusion (SD) (Rombach et al., 2022) shifts the denoising process into the latent space of a pre-trained autoencoder (Kingma & Welling, 2013), allowing for model scaling. Expanding from images to videos, diffusion models have seen success in text-to-video (T2V) generation (Wang et al., 2023; Chen et al., 2023; 2024; Zheng et al., 2024; Blattmann et al., 2023). In addition to the capacity to generate high-fidelity images based on text prompts, the T2V diffusion models implicitly learn the mapping from linguistic descriptions to video regions, providing an opportunity to repurpose them for RVOS. Among current T2V methods, ModelScope (Wang et al., 2023) and VideoCrafter (Chen et al., 2023; 2024) stand out for their open-source implementations, forming the backbone of our research.

**Visual-language Pre-training for Perception:** in addition to being highly effective in image and video generation, diffusion models have been shown to learn a strong representation of the natural image manifold. Several works have demonstrated that these representations can be re-purposed for classical computer vision problems, including semantic segmentation (Xu et al., 2023; Zhao et al., 2023; Zhang et al., 2023) and pixel-level correspondence (Tang et al., 2023), achieving an impressive degree of generalization. Others have shown that image diffusing models learn powerful representations of objects, enabling open-world novel view synthesis (Liu et al., 2023) and amodal segmentation (Ozguroglu et al., 2024). Most recently, Zhu et al. (2024) also leverages pretrained T2V models for RVOS however, our analysis shows that their approach fails to fully capitalize on the universal visual-language mapping learned in generative pre-training. In this work, we explore the application of video diffusion models to RVS, demonstrating how to maintain a high-level generalizability during fine-tuning.

In a separate line of work, visual-language representations learned with contrastive objectives (Bao et al., 2022; Radford et al., 2021) have been adapted for referring image (Lai et al., 2024; Rasheed et al., 2024; You et al., 2023; Xu et al., 2024) and video segmentation (Zhou et al., 2024). Although these models tend to be more light-weight, their performance remains limited, compared to both generative models, as well as classical referring segmentation approaches.

## 3 METHOD

### 3.1 LEARNING THE VISUAL-LANGUAGE MANIFOLD VIA VIDEO DENOISING

We build our REM upon T2V diffusion models (Wang et al., 2023; Chen et al., 2024; Zheng et al., 2024), which were originally designed to synthesize high-fidelity videos conditioned on language descriptions. To reduce the computational overhead, these models typically perform diffusion-denoising in the latent space, following Rombach et al. (2022). Concretely, given a video sequence $x$, a pretrained Variational Autoencoder (VAE) (Kingma & Welling, 2013) is used to project the video from pixel space to latent space: $\mathcal{E}(x) = z$; $\mathcal{D}(z) \approx x$, where $\mathcal{E}$ and $\mathcal{D}$ are the VAE encoder and decoder, respectively.

Considering the clean latent $z_0 \sim q(z_0)$, where $q(z_0)$ is the posterior distribution of $z_0$, video diffusion models progressively add Gaussian noise to $z_0$ during the *diffusion process*:

$$q(z_t|z_{t-1}) = \mathcal{N}(z_t; \sqrt{1 - \beta_t} z_{t-1}, \beta_t \mathbf{I}), \tag{1}$$

where $\beta_t$ is a variance schedule that controls the strength of the noise added in each timestep. The *denoising process* reverses this process, aiming to reconstruct the original latent. The estimated denoised latent at timestep $t - 1$ from $z_t$ is given by:

$$p_\theta(z_{t-1}|z_t) = \mathcal{N}(z_{t-1}; \mu_\theta(z_t, t), \mathbf{\Sigma}_\theta(z_t, t)), \tag{2}$$

where $\mu_\theta(z_t, t)$ and $\mathbf{\Sigma}_\theta(z_t, t)$ are the parameters of the Gaussian distribution, which are the targets of the diffusion model. The final denoising objective of video diffusion models is then:

$$\mathcal{L}_{\text{VDM}} := \mathbb{E}_{\mathcal{E}(x), \epsilon \sim \mathcal{N}(0,1), t} \left[ \| \epsilon - \epsilon_\theta (z_t, e_p, t) \|_2^2 \right]. \tag{3}$$

Language conditioning is integrated into diffusion models via the denoising network, $\epsilon_\theta(z_t, e_p, t)$. Concretely, a text encoder, such as CLIP (Radford et al., 2021), is used to tokenize the input prompt and generate the prompt embedding $e_p$. For UNet-based architectures, $e_p$ interacts with the latent representation through cross-attention modules, guiding the latent representations to generate diverse and semantically aligned videos based on text descriptions.

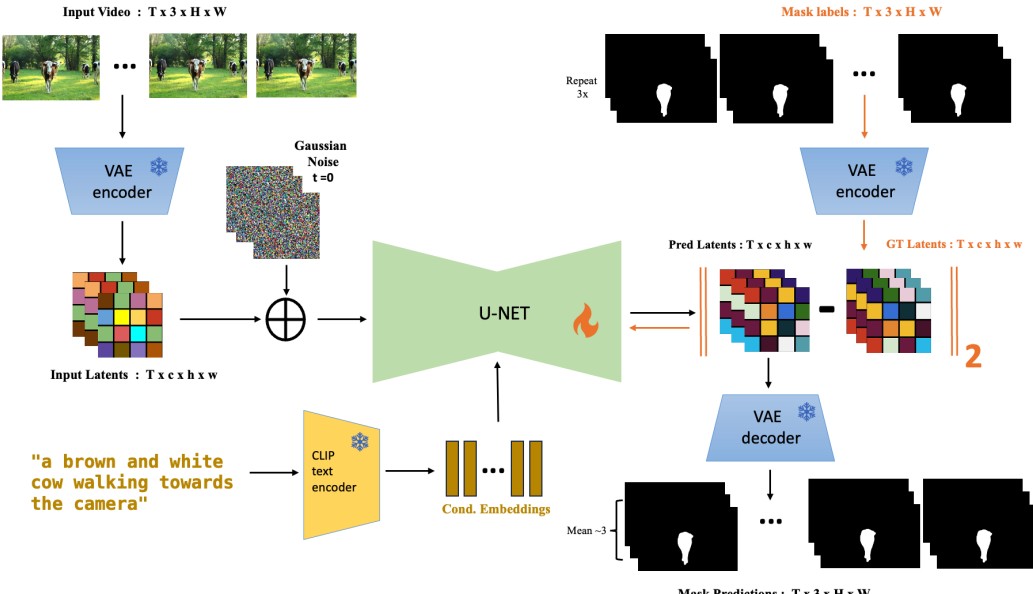

Figure 3: The model architecture of Refer Everything with Diffusion Models (REM). Like a video diffusion model it is based on, our approach takes video frames with added noise and a language expression as input. Our key insight is preserving as much of the diffusion representation intact as possible by supervising segmentation masks in the latent space of the VAE.

## 3.2 FROM LANGUAGE-CONDITIONED DENOISING TO REFERRING VIDEO SEGMENTATION

Referring Video Segmentation (RVS) is the task of segmenting an entity across space and time in a video based on a natural language expression. Formally, given a video sequence $x \in \mathbb{R}^{T \times 3 \times H \times W}$ and a text expression $p$, the goal is to produce a binary mask $m \in \mathbb{R}^{T \times H \times W}$, where true values indicate the presence of the referenced entity in each of the $T$ frames. This task aligns naturally with T2V diffusion models, as these models iteratively denoise latent representations during video generation, establishing a strong mapping between the entities described in the text and their corresponding regions in the video.

Several prior works have applied diffusion models to segmentation tasks (Zhao et al., 2023; Xu et al., 2023; Zhu et al., 2024), typically modifying the architecture to feed noisy latent representations and text embeddings into the denoising UNet, extracting intermediate features for downstream tasks. These approaches often employ task-specific decoders in a conventional discriminative learning setup, effectively repurposing diffusion models as feature extractors. However, this extensive architectural modification results in divergence from the pretraining phase when fine-tuning on narrow-domain datasets. This in turn causes the model to lose much of the general knowledge acquired during pretraining, which is crucial for robust generalization.

In our approach, rather than using diffusion models solely as feature extractors, we preserve the original architecture and specifically adapt it for the RVS task. This enables us to fully leverage the extensive knowledge encoded in diffusion models while refining them to meet task-specific requirements, without discarding critical information. As illustrated in Figure 3, REM re-purposes the denoising network by shifting its objective from predicting noise to predicting mask latents. This subtle yet powerful adaptation allows the model to retain its pretraining knowledge while enhancing its ability to tackle video segmentation. The technical details of this adaptation are elaborated below.

## 3.3 REFER EVERYTHING WITH DIFFUSION MODELS

Instead of learning a mask decoder from scratch, we reuse the VAE from video diffusion models. To adapt the target masks to the VAE, we broadcast the single-channel mask into three channels by repeating the mask. For simplicity, we still denote this three-channel mask sequence as $m$. The pretrained VAE can then map the mask sequence into the latent space: $\mathcal{E}(m) = z^m$ and $\mathcal{D}(z^m) \approx m$.

**Training and Optimization.** Starting with the clean video latent $z_0$, the first step remains the same – applying noise to shift it into a noisy distribution, following Equation 1, as the denoising network operates in the noisy latent space with the corresponding timestep embeddings as input. However, since our objective has shifted from denoising to mask prediction, we prioritize using latents that remain as clean as possible. Therefore, we always set the noisy timestep to its minimum value, $t = 0$, which shifts $z_0$ to $z_1$. Next, we input the video latent $z_1$, the prompt embedding $e_p$, and the timestep $t = 0$ into the denoising network $\epsilon_\theta$. We then supervise the predicted mask latents using an $\mathcal{L}_2$ loss:

$$\mathcal{L}_{\text{REM}} = ||z^m - \epsilon_\theta(z_1, t_0, e_p)||^2. \tag{4}$$

**Model Inference.** During inference, we follow the same procedure, but decode directly from the predicted mask latent to generate the three-channel mask predictions: $\hat{m} = \mathcal{D}(\epsilon_\theta(z_1, t_0, e_p))$. We then compute the single-channel masks by averaging the three-channel masks pixel-wise, and apply a constant threshold of 0.5 to binarize the masks.

## 4 BENCHMARK DESIGN AND COLLECTION

In this section, we discuss our approach to collecting a new benchmark that would expand the focus of Referring Video Segmentation outside the domain of object tracking. As covering the entire spectrum of concepts that can be spoken of in videos would be extremely costly, we seek to identify a subset of the problem that requires joint modeling of language and temporal dynamics. To this end, we choose to focus on dynamic processes, which we define as temporally evolving *events*, where the subjects undergo continuous changes in state, shape, or appearance. Crucially, the subjects in this context are not limited to objects, but include all concepts that are spatio-temporally localizable in videos, such as light or fire. The key steps for collecting this new benchmark, which we call Referring Video Process Segmentation (Ref-VPS), include selecting representative videos and annotating them with referring expressions and segmentation masks.

### 4.1 VIDEO SELECTION

To source the videos for our benchmark we require a large, public and diverse database that is queriable with natural language and allows re-distribution of content for research purposes. Based on these requirements, we choose the TikTok social media platform which has over 1 billion active users across the world and receives tens of millions of video uploads daily, capturing a wide range of dynamic visual content. TikTok's policies generally allow for free redistribution of content, with individual users having the option to opt out.

To search for videos that capture dynamic processes, as defined above, we first identify a non-exhaustive list of six broad and possibly overlapping concepts (e.g. 'object transformations', or 'entities with dynamic boundaries', full list together with definitions provided in Section A in the appendix). Then, for each concept we ask ChatGPT (OpenAI, 2023) to provide a list of concrete examples together with multiple text queries for search on TikTok (*e.g.*, 'a wax candle melting' for 'object transformations'), resulting in 120 individual concepts. We retrieve over a 1000 samples based on these queries; however a majority of the queries did not yield suitable videos because of the physical nature of the event (e.g. events like 'soil erosion' are not typically captured on TikTok due to their long temporal span), or ambiguity of search query not lending itself to being accurately captured on TikTok. After removing irrelevant videos, the retrieved set is reduced to 342 samples.

We then manually filter these videos based on the following criteria: (1) videos that do not feature significant dynamic changes of the subject (*e.g.*, mostly stationary clouds in the sky); (2) dynamic processes that occur too rapidly to allow for the labeling of a sufficient number of non-empty frames (*e.g.*, flashes of lightning); (3) video with frequent shot changes, which make it impossible to extract an interrupted clip capturing the event of interest. Additionally, for videos that represent compilations of similar events, we split them into individual clips and treat each one independently. The resulting dataset contains 111 video clips representing 38 dynamic process concepts. The entire dataset is intended for 0-shot evaluation, so we do not define any additional splits. A representative sample of the videos is shown in Figure 1.

## 4.2 ANNOTATION COLLECTION AND EVALUATION

To label the videos selected above, we begin by adjusting the temporal boundaries of each clip to focus on the event of interest and avoid shot changes. We also make sure that the event is captured in its entirety whenever possible, including some context before and after it. The clips are then exported at 24 FPS as image frames. If a video contains irrelevant frames, such as the TikTok logo at the end, we crop the frames accordingly to remove the padding.

To collect referring expressions, we first manually identify the entity of interest in each clip. The selected entity is then labeled with referring expressions by two independent annotators. Each annotator provides two expressions for the target, resulting in a total of four expressions per clip, capturing different ways to describe the same phenomenon. Following the standard protocol (Khoreva et al., 2019; Seo et al., 2020), models are evaluated on all queries and the results are averaged.

Finally, we densely label the targets identified above with segmentation masks at 24 FPS. To this end, we employ a semi-automatic pipeline, capitalizing on the recently introduced SAM2 (Ravi et al., 2024) foundational model for interactive video segmentation. In particular, we provide positive and negative click annotation in the middle frame of a video first to ensure accurate boundary segmentation. SAM2 then automatically segments the entity of interest in the frame, as well as propagates the mask across the entire clip. We interactively improve segmentation quality by providing additional clicks as needed. In the end, we manually refine the masks in frames where SAM2 fails and label ambiguous regions as Ignore. A visualizations of Ref-VPS annotations together with additional statistics of our benchmark are included in the appendix. For evaluation, we follow Tokmakov et al. (2023) and only report region similarity $\mathcal{J}$ as contour accuracy $\mathcal{F}$ is often not well defined for the entities like smoke or light which are frequent in Ref-VPS. Pixels inside the Ignore regions are not included in the metric calculation.

## 5 EXPERIMENTS

**Datasets and Evaluation.** We use the popular RVOS benchmarks Ref-YTB (Seo et al., 2020) and Ref-DAVIS (Khoreva et al., 2019) for evaluating our model's performance on object tracking. Ref-YTB contains 3978 videos with 15k referring expressions and spans 94 common object categories. Ref-DAVIS (Khoreva et al., 2019) contains 90 videos and is evaluated 0-shot (similarly to other methods) using the official evaluation code. For Ref-YTB (Seo et al., 2020) we use their public labels for training and the evaluation is done on the official challenge server. Following standard practice, in addition to Ref-YTB, we use an image segmentation dataset Ref-COCO (Yu et al., 2016) for training, which across all three versions has 320k image-text samples.

For evaluating generalization to rare objects and 'Stuff' categories, we use the BURST (Athar et al., 2023) and VSPW (Miao et al., 2021) datasets respectively. For more details about these benchmarks please refer to Section C.1 in the Appendix. Finally, we evaluate REM and the strongest baselines on our newly introduced Ref-VPS benchmark that focuses on dynamic process segmentation (detailed in Section 4), and contains 111 videos across 38 concepts. All these datasets are only used for evaluation (*i.e.*, the results are zero-shot).

For Ref-YTB (Seo et al., 2020) and Ref-DAVIS(Khoreva et al., 2019) we use the standard evaluation metrics - Region Similarity ($\mathcal{J}$), Contour accuracy ($\mathcal{F}$) and their mean ($\mathcal{J}\&\mathcal{F}$). For all other evaluations we use the Region Similarity ($\mathcal{J}$) metric.

## 5.1 REFERRING VIDEO OBJECT SEGMENTATION RESULTS

In this section we compare REM to the state of the art on the standard RVOS benchmarks. We report results on the validation set of Ref-DAVIS (Khoreva et al., 2019) and the test set of Ref-YTB (Seo et al., 2020) in Table 1. Our method outperforms the state of the art on all metrics on Ref-DAVIS and is only second to UNINEXT (Yan et al., 2023) on Ref-YTB. Note that this approach is specifically designed for object segmentation and utilizes more than 10 datasets with localization annotations like bounding boxes and masks for training. In contrast, REM adopts an architecture of a video generation model and is only fine-tuned on one image and one video segmentation dataset. Despite this, our method is competitive with UNINEXT on standard RVOS benchmarks, and as we will show next, outperforms it out-of-domain by up to 46% in terms of Region Similarity.

| Method | Pretraining Data | Mask/Box Supervision | Ref-DAVIS | | | Ref-YTB | | |
|---|---|---|---|---|---|---|---|---|
| | | | $\mathcal{J}\&\mathcal{F}$ | $\mathcal{J}$ | $\mathcal{F}$ | $\mathcal{J}\&\mathcal{F}$ | $\mathcal{J}$ | $\mathcal{F}$ |
| Referformer (Wu et al., 2022a) | ImageNet + Kinetics + SSv2 | Ref-COCO/+/g + Ref-YTB | 61.1 | 58.1 | 64.1 | 62.9 | 61.3 | 64.6 |
| MUTR (Yan et al., 2024) | ImageNet + Kinetics + SSv2 | Ref-YTB + AVS | 68.0 | 64.8 | 71.3 | 68.4 | 66.4 | 70.4 |
| VLMO-L (Zhou et al., 2024) | Unknown | Ref-COCO/+/g + Ref-YTB | 70.2 | 66.3 | 74.1 | 67.6 | 65.3 | 69.8 |
| UNINEXT (Yan et al., 2023) | Object365 | 10+ Image/Video datasets | 72.5 | 68.2 | **76.8** | **70.1** | **67.6** | **72.7** |
| VDIT (Zhu et al., 2024) | LAION5B+WebVid | Ref-COCO/+/g + Ref-YTB | 69.4 | 66.2 | 72.6 | 66.5 | 64.4 | 68.5 |
| REM (Ours) | LAION5B+WebVid | Ref-COCO/+/g + Ref-YTB | **72.6** | **69.9** | 75.29 | 68.4 | 67.05 | 69.73 |

Table 1: Comparison to the state of the art on the validation set of the Ref-DAVIS and the test set of Ref-YTB benchmarks using the standard metrics. Our method performs on par with the strong UNINEXT approach, despite not being specifically designed for object localization and having access to only a fraction of the localization labels used by that method.

| Method | MUTR | UNINEXT | VDIT | REM (Ours) |
|---|---|---|---|---|
| VSPW | 10.5 | 10.1 | 12.7 | **15.2** |
| BURST | 27.9 | 30.2 | 30.9 | **40.4** |

Table 2: $\mathcal{J}$ Comparison to the state of the art on the 'Stuff' categories in the val set of VSPW and on the joint val and test sets of BURST. Our approach demonstrates much stronger generalization, notably, outperforming VDIT which is based on the same diffusion backbone.

| Method | MUTR | UNINEXT | VDIT | REM (Ours) |
|---|---|---|---|---|
| Ref-VPS ($\mathcal{J}$) | 24.07 | 26.25 | 35.27 | **48.96** |

Table 3: Comparison to the state of the art on our new Ref-VPS benchmark. REM shows much stronger generalization to challenging, dynamic concepts in this dataset compared to the baselines by effectively capitalizing on Internet-scale visual-language pre-training.

Another notable observation is that REM also outperforms VDIT (Zhu et al., 2024), which is built on top of the same video diffusion backbone of Wang et al. (2023), on both datasets. This result demonstrates the effectiveness of our approach preserving the visual-language representations learned on the Internet data, which will become even more evident in out-of-domain evaluation.

## 5.2 OUT-OF-DOMAIN GENERALIZATION

Before analyzing referring video segmentation methods on our new Ref-VPS benchmark, we report a preliminary generalization study on existing open-world tracking BURST dataset (Athar et al., 2023) as well as on the 'Stuff' categories (Caesar et al., 2018) from VSPW (Miao et al., 2021) in Table 2. BURST is an open-world video object segmentation benchmark featuring larger object diversity that the standard RVOS benchmarks, whereas VSPW tests the ability to generalize to non-object categories. We report results on the validation set of VSPW and combined validation and test sets of BURST and compare to the top performing methods from Table 1 that have public models. All the evaluations reported in this section are zero-shot.

Firstly, we observe that on both out-of-domain challenges our method outperforms all the baselines by significant margins. The improvements are especially noticeable on BURST, demonstrating that our method successfully preserve the strong object representation learned by Internet-scale pre-training of the diffusion backbone. In contrast, VDIT looses this generalization capacity during fine-tuning and only performs on par with UNINEXT. On the 'Stuff' categories all the methods do relatively poorly, reflecting the challenge of generalizing to more amorphous 'Stuff'. Here VDIT maintains a lead over entirely object-centric UNINEXT but REM still outperforms both baselines.

Finally we compare REM to the top-performing RVS baselines on our new Ref-VPS benchmark in Table 3. Here the differences between the methods are a lot more pronounced on this benchmark compared to other datasets, highlighting the value of our benchmark in assessing video-language understanding capabilities of neural representations. Our approach outperforms all baselines by up to 28% in Region Similarity, and notably surpasses the top RVOS method, UNINEXT, by 46%. While generative pre-training enhances VDIT's generalization ability over UNINEXT, it struggles to preserve its representations as effectively as our method.

A qualitative comparison of REM with VDIT and UNINEXT on Ref-VPS is provided in Figure 4. We can see that both baselines exhibit object-centric bias, as in the examples with the lizard skin in row 1 and blue smoke in row 5. While VDIT show better generalization to non-object concepts (e.g. in row 2), it often simply segments the dominant region in the video (see the last row in Figure 4). In contrast, REM shows both good coverage of the rare concepts and high precision with respect to the language prompt. See more examples of highly dynamic sequences in Section B.1 in the appendix.

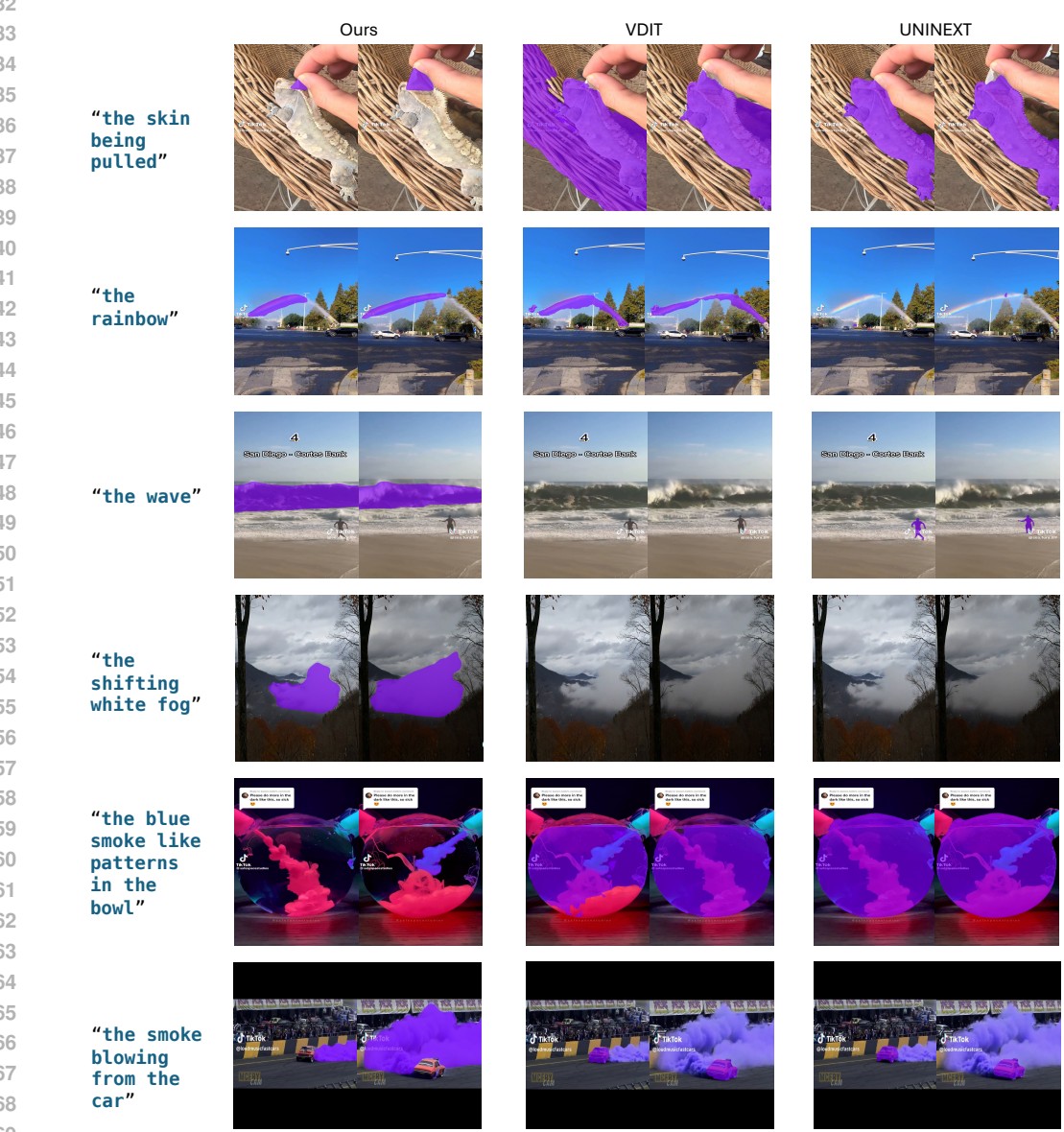

Figure 4: Qualitative results of REM and state of the art baselines on our Ref-VPS benchmark. Our method demonstrates both superior coverage of rare, dynamic concepts and higher segmentation precision. Video comparisons are available here.

## 5.3 ABLATION ANALYSIS

In this section, we analyze our proposed approach of transferring generative representations to the task of RVS. We report results on one representative RVOS benchmark (Ref-YTB) and on our new Ref-VPS. Note that for efficiency we fine-tune all the models on a subset of image and video data (12000 samples) so the results are lower than those reported in the previous section.

**Generative pre-training.** We begin by evaluating the effect of the generative pre-training strategy in Table 4. Firstly, we design a frame-level baseline which fine-tunes StableDiffusion (Blattmann et al., 2023) on every frame individually (row 1 in the table). While this variant has no temporal modeling capacity, its architecture is similar to UNINEXT (Yan et al., 2023) - the state-of-the-art approach for RVOS. Interestingly, it strongly under-performs compared to our best video-based variant not only on our Ref-VPS but also on the object-centric Ref-YTB benchmark. These results demonstrate

Table 4: Analysis of the effects of generative pre-training and discriminate fine-tuning strategies on Ref-YTB and Ref-VPS. The key to success of REM is capitalizing on Internet-scale image and video pre-training and preserving as much of this representation as possible.

| Backbone | Decoder | Ref-YTB ($\mathcal{J}\&\mathcal{F}$) | Ref-VPS $\mathcal{J}$ |
|---|---|---|---|
| Stable Diffusion 2.1 | Frozen VAE | 59.38 | 28.36 |
| VideoCrafter-1 | | 59.10 | 27.15 |
| VideoCrafter-2 | | **65.00** | 35.66 |
| ModelScope T2V | | **64.57** | **37.80** |
| ModelScope T2V | CNN | 60.47 | 25.09 |
| ModelScope T2V | MLP | 59.35 | 31.75 |

that, despite the fact that images are the dominant data source in generative pre-training, fine-tuning StableDiffusion for video generation is crucial for learning an effective representation for tracking.

Next, we compare several strategies for learning video diffusion models. We begin by studying two variants of the VideoCrafter model (Chen et al., 2023; 2024) (denoted as VideoCrafter-1 and VideoCrafter-2 in Table 4). They are both trained on 600M images from LAION (Schuhmann et al., 2022) and 10-20M Internet videos. However, VideoCrafter-2 is further tuned to increase the quality of the generated samples. Our findings indicate that this fine-tuning step leads to significant performance gains across both benchmarks. This suggests that improving the quality of video generation models can directly translate to enhanced performance in our video segmentation framework.

Finally, we evaluate ModelScope (Wang et al., 2023), which is trained on larger LAION 2B and a comparable amount of video samples (last row in Table 4). This model delivers performance comparable to the best version of VideoCrafter on the Ref-YTB benchmark, while demonstrating superior generalization to more challenging concepts in Ref-VPS. These results further highlight that both large-scale pre-training on image data as well as learning to model video-language interactions are crucial components for robust RVS representation learning.

**Fine-tuning strategy.** We now ablate the effectiveness of our design decision to re-use a frozen VAE decoder for mask prediction, rather than replacing it with a dedicated mask prediction module, as was done in some of the prior work (Zhao et al., 2023; Zhu et al., 2024). To this end, we replace the VAE with a CNN mask decoder adopted from (Zhao et al., 2023), as well as with an MLP adopted from SegFormer (Xie et al., 2021), and train it jointly with the rest of the model (last two row in Table 4). Removing the pre-trained VAE decoder has a moderate negative effect on performance on Ref-YTB, but, notably, destroys the model's ability to generalize our challenging Ref-VPS benchmark. This result underscores the main message of our paper - preserving as much of the representation learned during generative pre-training is key for achieving generalization in referring video segmentation.

# 6 DISCUSSION

In this paper, we proposed REM, a framework that capitalizes on Internet-scale video-language representations learned by diffusion models to segment a wide range of concepts in video that can be described through natural language. Our key insight is that changing as little as possible in the representation is key to preserving its universal mapping between language and visual concepts during fine-tuning. To illustrate the benefits of our approach, we have also collected Ref-VPS - a new benchmark for referring segmentation of dynamic processes in videos, which significantly expands the scope of existing RVOS datasets. Our extensive experimental evaluation demonstrates that, despite only being trained on object masks, REM successfully generalizes to highly dynamic concepts in Ref-VPS, outperforming all prior work by up to 28% .

Despite REM's impressive generalization abilities, the problem of RVS is far from being solved. In Figure D in the Appendix we visualize a few failure cases of our method. REM still exhibits some object-centric bias and struggles with extremely fast processes. Exploring ways to preserve even more of the representation learned during generative pre-training, e.g. via low-rank adaptation (Hu et al., 2022) of the visual backbone, is a very promising direction to address some of these issues. In addition, note that REM should be seen as a generic framework where the backbone of Wang et al. (2023) can be easily replaced with a more advanced representation, tracing the progress of language-conditioned video generative models.

## CODE OF ETHICS

There is no obvious negative societal impact from our work. The potential negative impact is likely the same as other research on large-scale generative models with the legal concern on the training data.

## REPRODUCIBILITY STATEMENT

We provide extensive descriptions of the implementation details in the appendix. Also, we will release the code upon acceptance.

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

| | |
|---|---|
| Clips | 111 |
| Frames | 14452 |
| Concepts | 38 |
| Avg len. (s) | 5.47 |
| Ann FPS | 24 |
| Min-resolution | $712 \times 576$ |
| Max-resolution | $1024 \times 576$ |

Table A: Statistics of our Ref-VPS benchmark.

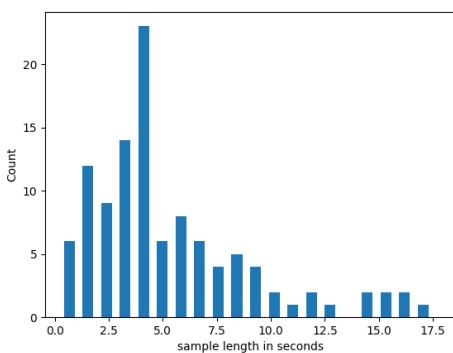

Figure A: Distribution of sample lengths in Ref-VPS

## A    DATASET DETAILS

### A.1    DATASET COLLECTION

We use the following categories of dynamic processes to collect videos for our Ref-VPS benchmark:

- Temporal object changes: Concepts involving changes over time (e.g., object deformation, melting)

- Motion Patterns: Concepts involving movement and displacement of non-object regions (e.g., water ripples, flickering flames)

- Dynamic environmental changes: Changes in the environment that affect spatial regions over time (e.g. clouds moving across the sky, waves rising )

- Interaction Sequences: Concepts involving interactions between objects (e.g., bullet hitting glass, object collisions)

- Pattern evolution: Concepts where patterns or textures evolve or change dynamically (changing patterns of smoke dispersion, fluctuating light levels)

The distribution of our sample lengths can be seen in Figure  A. Most of our samples are around 2.5 to 5 secs in length but can go up to 17 seconds. A comprehensive list of key statistics can be found in Table.  A.

### A.2    ANNOTATION VISUALIZATIONS

We show a sample of Ref-VPS segmentation mask annotations Figure B. Our annotations are accurate, with the entire extent of the wave labeled in the third row, and the entire icicle in the second row. Rows 1 and 4 illustrate handling of ambiguous scenarios, where only the confident regions of the glowing water and of the light column are labeled as target, and the ambiguous regions are labeled as Ignore (shown in gray). Pixels inside the Ignore regions are not included in the metric calculation. This approach ensures that the metrics focus on evaluating the most reliable regions of the masks, avoiding arbitrary penalties for ambiguous boundaries.

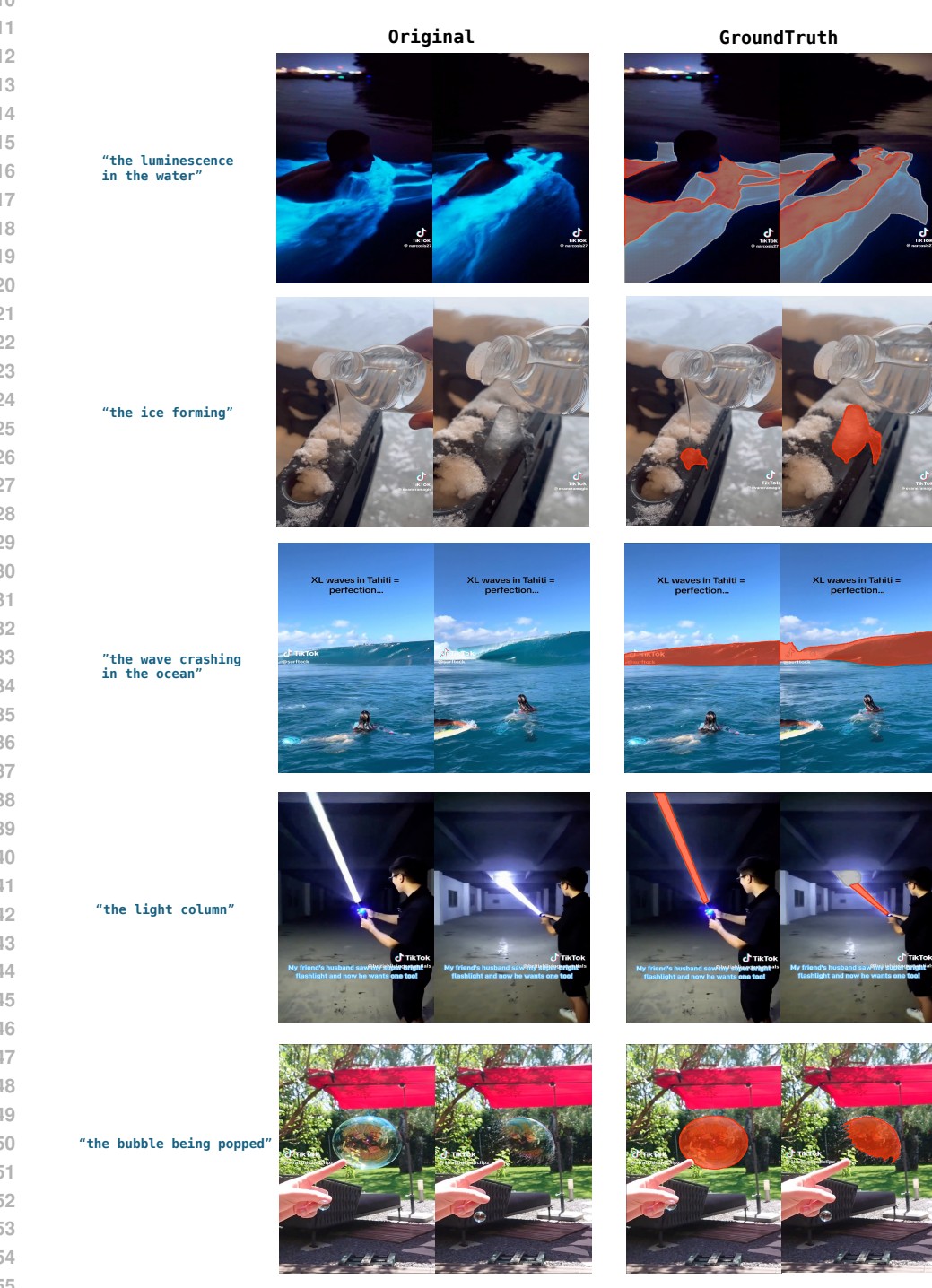

Figure B: Samples from our Ref-VPS dataset. Ground-truth masks are shown in red and the Ignore regions are shown in gray. Pixels inside the Ignore regions are not included in the metric calculation.

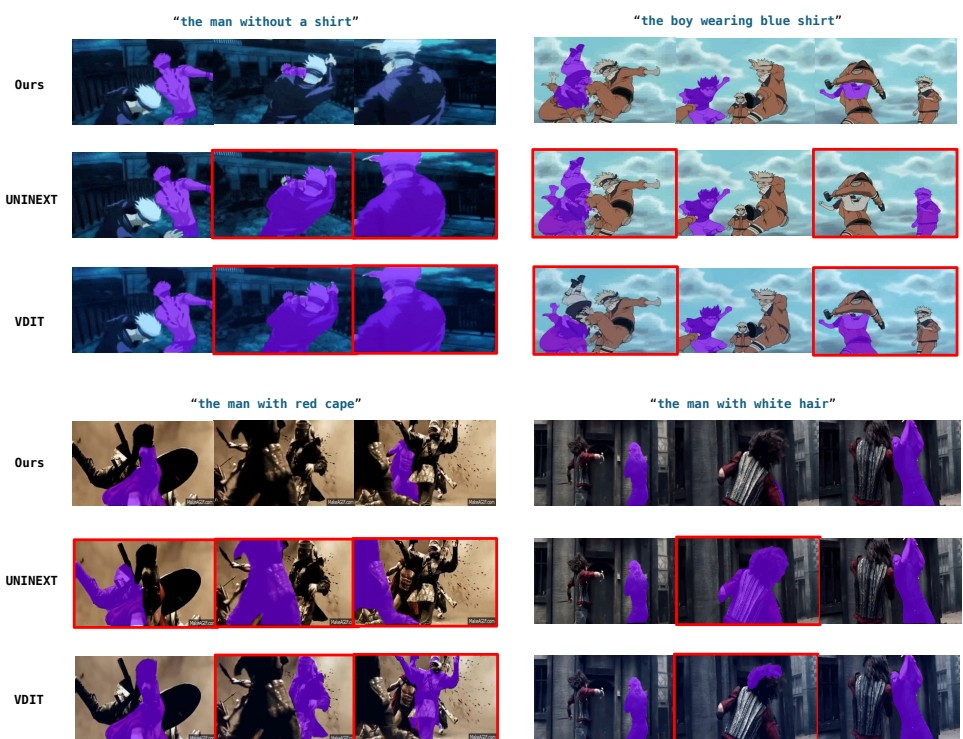

Figure C: Qualitative comparison of REM with state-of-the-art baselines on dynamic and challenging fight scenes. The incorrectly labeled frames are outlined in red. Our method is way better at handling frequent occlusions and POV changes. For better illustration of the differences, please watch the full videos here.

## B    ADDITIONAL EXPERIMENTAL EVALUATIONS

### B.1    EVALUATION ON CHALLENGING FIGHT SCENES

Fight sequences in movies, television and animated shows pose a unique set of challenges. Typically fight scenes are characterized by objects/characters undergoing severe and frequent occlusions and leaving the frame entirely, coupled with frequent pose changes of the camera. This leads to drastic changes in the appearance of the object and requires high levels of temporal and semantic consistency to accurately track, re-identify, and segment the referred entity. Our diffusion fine-tuning method excels in this domain of super challenging samples as illustrated in Figure C. We can clearly see that UNINEXT and VDIT both fail whenever there is a large occlusion causing the referred entity to become invisible. Even though VDIT uses Video diffusion features, their method is unable to leverage the temporal consistency learned during Video Diffusion pre-training as well our method. For a more illustrative comparison, we highly recommend you watch the full videos linked in the caption of Figure C.

### B.2    FAILURE CASES

A few representative failure cases of REM on Ref-VPS are shown in Figure D. Our method suffers from object-centric bias in the most challenging scenarios and struggles with extremely fast processes.

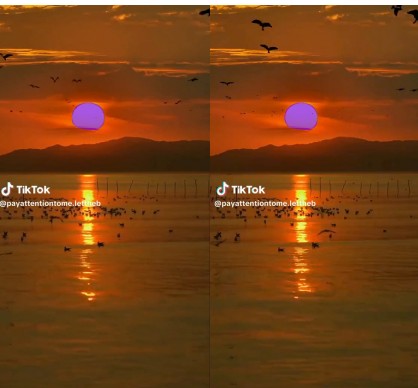

"the light reflecting off the bald head"    "the light reflected in the water"

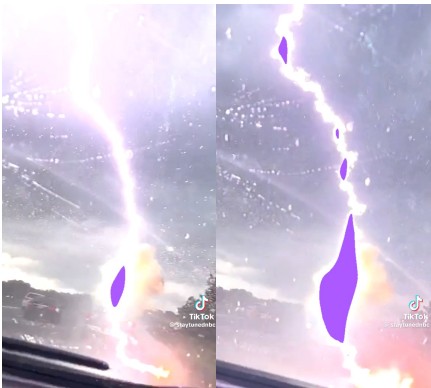

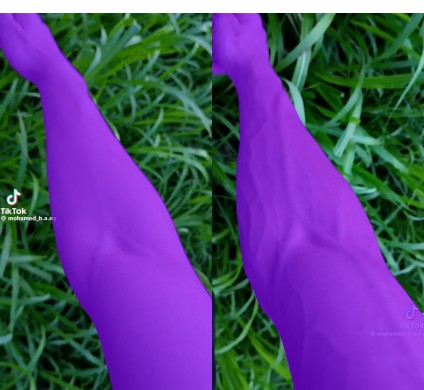

"the lightning strike"    "the veins on the arm"

Figure D: Some failure cases of our REM on Ref-VPS. The model still exhibits some object-centric bias and struggles with extremely dynamic entities like the lightning.

### B.3    CONCEPT COVERAGE PLOT ON BURST

In case of generalization to object concepts on BURST, our method outperforms the next best by at least 9.5 %. In Figure E we show that our method has a much better coverage of different object concepts compared to other methods. We are especially better in the long-tail region of object concepts as illustrated in the figure.

### B.4    TEMPORAL CONSISTENCY EVALUATION

Accurately evaluating the temporal consistency of video segmentation methods is notoriously challenging, because it is hard to distinguish between predicted mask changes that are due to the method's inconsistency and the changes that are due to the true target deformations. Notably, the temporal consistency metric proposed in the original DAVIS dataset (Pont-Tuset et al., 2017) was only applied to videos with no significant object deformations and no occlusions and was eventually phased out by the dataset's authors.

Recognizing these limitations, we implemented a straightforward consistency metric by computing the average difference of IoU between the model's prediction and the ground truth mask in consecutive frames. Formally,

$$\text{Temp. Con.} = \frac{1}{N}\sum_{n=1}^{N}\left[\frac{1}{T_n}\sum_{t=1}^{T_n}(IoU(Pred_{t+1}, GT_{t+1}) - IoU(Pred_t, GT_t))\right], \quad (5)$$

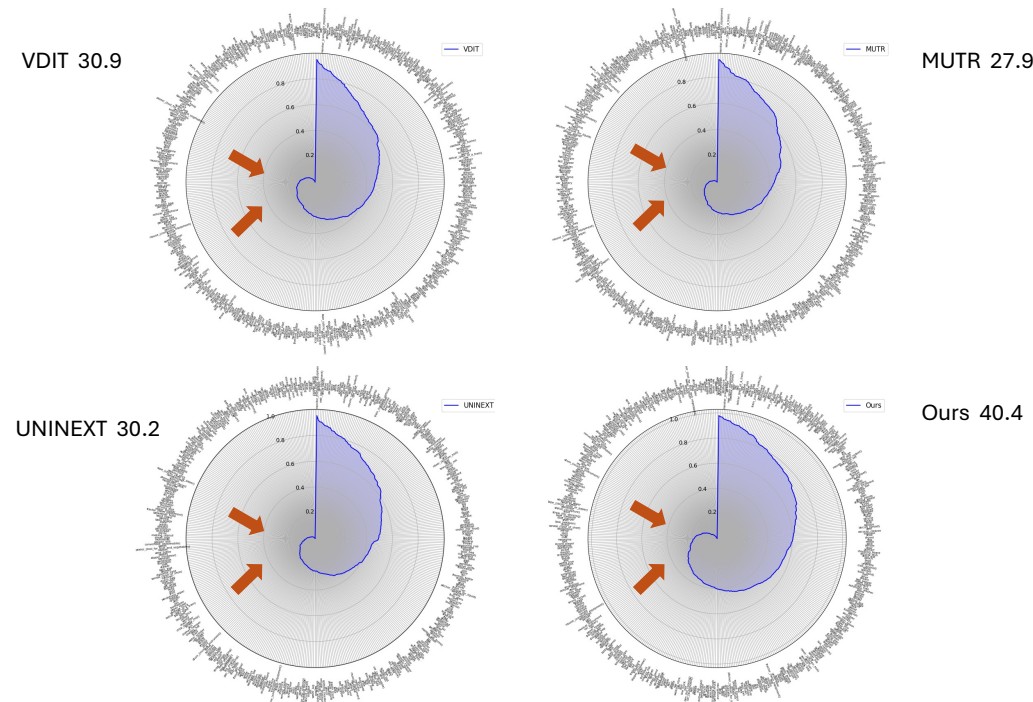

Figure E: Class-wise J scores demonstrating concept coverage on BURST. As indicated by the arrows, we are much better in the long-tail region compared to other methods.

| Method | Ref-VPS | | Ref-DAVIS | |
|---|---|---|---|---|
| | $\mathcal{J}$ | Temp. Con. | $\mathcal{J}$ | Temp. Con. |
| MUTR | 24.1 | 2.9 | 64.8 | 3.4 |
| UNINEXT | 26.3 | 5.2 | 68.2 | 5.2 |
| VDIT | 35.3 | 4.7 | 66.2 | 3.1 |
| REM (Ours) | 49.0 | 2.8 | 69.9 | 2.1 |

Table B: Temporal Consistency comparison to the state of the art on Ref-VPS and Ref-DAVIS. Our approach demonstrates the best temporal consistency on both object-centric and non-object-centric datasets.

where $N$ is the number of samples and $T_n$ is the number of frames in the $n^{th}$ sample. Lower numbers indicate better temporal consistency on this metric, and it is easy to see that simply outputting empty masks would result in the perfect consistency score of 0. Hence, as with any temporal consistency metric, it should always be considered jointly with a prediction accuracy metric.

We report region similarity and temporal consistency on Ref-VPS and Ref-DAVIS (the two datasets that extract frames at 24 fps) in Table B. The results demonstrate the superior temporal consistency of REM on both object-centric and non-object-centric datasets. Notably, UNINEXT - the state-of-the-art RVOS approach, shows the worst temporal consistency out of all methods. MUTR achieves a strong temporal consistency score on Ref-VPS precisely because it often outputs empty predictions, as can be seen from its low region similarity score.

## B.5 COMPARISONS ON AMBIGUOUS OR OVERLAPPING SCENARIOS

To understand how well our method handles ambiguous scenarios in Ref-VPS, we add a visual comparison between REM and VDIT, the strongest baseline on this benchmark, in Figure F. It is clear to see that, although for many of these samples, no perfect prediction exists, the outputs of our model are both more accurate in the confident regions and more consistent. For example, in the first row, our method only segments the clearly visible regions of lava once it is hit by a wave, whereas VDIT segments the entire wave as well. In the second row, REM consistently segments all the glowing water, whereas VDIT only covers a few patches.

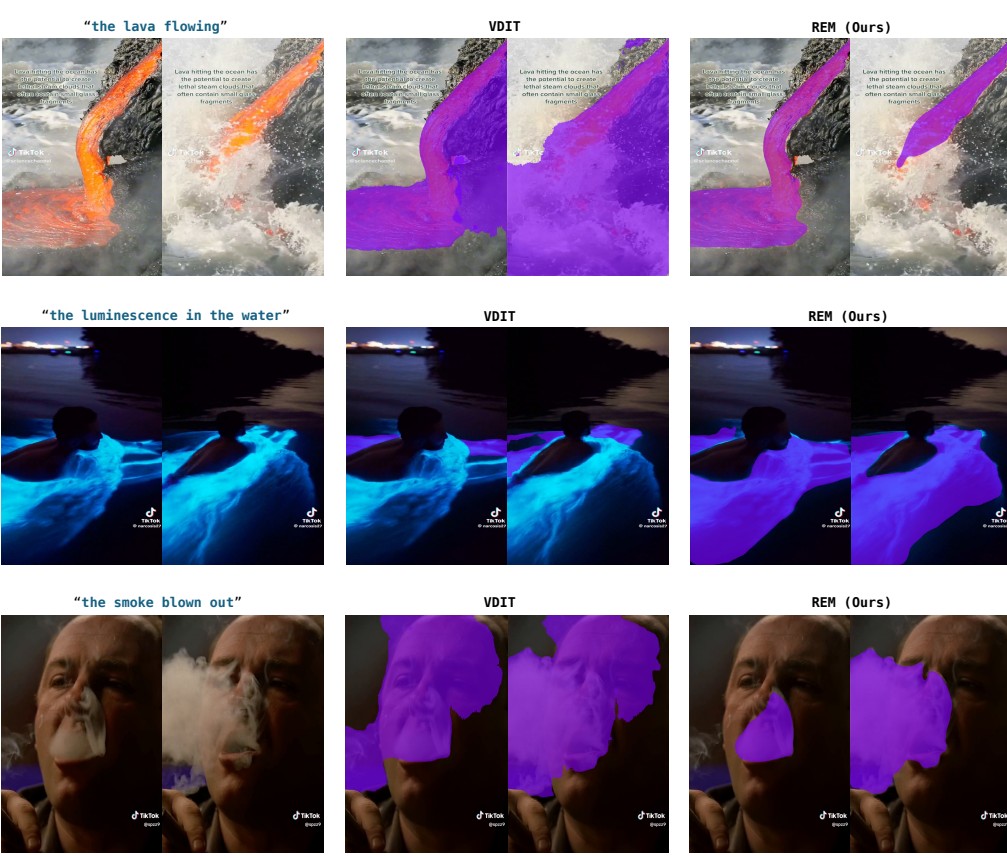

Figure F: Comparison on ambiguous or overlapping scenarios in Ref-VPS between VDIT and REM (Ours). While no single perfect prediction exists for these samples, our method is both more precise and more consistent.

### B.6 COMPUTATIONAL COST

We report the inference speed and the memory consumption of REM alongside the primary baselines from our paper, using their public implementations in Table C. These values are estimated under the following protocol: inference was performed on 32-frame clips from Ref-DAVIS on a single A100 GPU with averages computed over 80 runs. As shown, the inference costs of REM align with those of other state-of-the-art approaches.

For training, REM takes 174 hours on 4 A100 GPUs. Other methods do not report their training costs, so we have estimated them ourselves (excluding i/o time) given the same computational budget, and report the results together with the memory consumption per GPU in Table D. Our costs are on par with most prior works. Notably, UNINEXT – the state-of-the-art RVOS approach, takes 6.3 times longer to train than REM, since it utilizes more than 10 datasets with object supervision to achieve top object segmentation results. In contrast, REM effectively capitalizes on Internet-scale pre-training, allowing it to achieve competitive performance with UNINEXT on the traditional RVOS dataset and significantly outperform it out-of-domain. All at a fraction of the training cost.

| Method | Memory (GB) | Speed (FPS) |
|--------|-------------|-------------|
| MUTR | 34.1 | 13.6 |
| UNINEXT | 9.7 | 3.3 |
| VDIT | 72.8 | 7.1 |
| REM (Ours) | 41.8 | 7.1 |

| Method | Memory (GB) | Total Runtime (hr) |
|--------|-------------|--------------------|
| MUTR | 30.4 | 134 |
| UNINEXT | 30.2 | 1906 |
| VDIT | 68.5 | 260 |
| REM (Ours) | 61.8 | 174 |

Table C: Inference costs of REM and top RVS methods on Ref-DAVIS. Both the memory requirements and the runtime of REM are on par with other models in the literature.

Table D: Training costs of REM and top RVS methods. Our costs are on par with prior work and are notably significantly lower compared to UNINEXT – the state-of-the-art RVOS approach.

| Method | Mask/Box annotations | Ref-Davis ($\mathcal{J}\&\mathcal{F}$) | Ref-YTB ($\mathcal{J}\&\mathcal{F}$) |
|--------|----------------------|-----------|----------|
| Referformer | RefCOCO/g/+, Ref-Youtube-VOS | 61.1 | 62.9 |
| MUTR | Ref-Youtube-VOS, AVS | 68.0 | 68.4 |
| VLMO-L | RefCOCO/g/+, Ref-Youtube-VOS | 70.2 | 67.6 |
| UNINEXT | Objects365, COCO, RefCOCO/g/+, GOT-10K, LaSOT, TrackingNet, Youtube-VOS, BDD100K, VIS19, OVIS, Ref-Youtube-VOS | 72.5 | **70.1** |
| VDIT | RefCOCO/g/+, Ref-Youtube-VOS | 69.4 | 66.5 |
| REM (Ours) | RefCOCO/g/+, Ref-Youtube-VOS | **72.6** | 68.4 |

Table E: Comprehensive list of bounding/mask supervision used by all methods.

## C IMPLEMENTATION DETAILS

### C.1 EVALUATION BENCHMARKS

Neither BURST (Athar et al., 2023) nor VSPW (Miao et al., 2021) contains referral text for the segmented entities. Since we want to strictly evaluate the entity recognition capacity of the models, we automatically generate referral expressions using only the category of the masked entity as "the <class>" (e.g. "the hat"). For VSPW we evaluate the validation set which has 66 different stuff categories. In the case of BURST, the validation and test sets contain object categories that the other split does not. So here we evaluate the combined validation and test set which contains 454 classes and a total of 2049 sequences.

### C.2 TRAINING DETAILS

We train our model using 4 NVIDIA 80GB A100 GPUs. We use ModelScope T2V (Wang et al., 2023) as our base video diffusion architecture and set the input noise level to 0. In the first stage, we fine-tune only the spatial weights using image-text samples from Ref-COCO (Yu et al., 2016) for 1 epoch and then fine-tune all weights for 40 epoch using Ref-YTB (Seo et al., 2020) video-text samples and 12k samples from Ref-COCO jointly. In the second stage, the image samples from Ref-COCO are converted to pseudo videos through augmentations following Referformer (Wu et al., 2022b). We freeze the CLIP text encoder and the VAE encoder and decoder during training and only fine-tune the U-Net. We use a low learning rate of 1e-6 in both stages and the AdamW optimizer (Loshchilov et al., 2019). The number of frames T is set to 8 during training and 72 during evaluation.

