# OpenReview forum: "ReferEverything: Towards segmenting everything we can speak of in videos"
_ICLR.cc/2025/Conference — Submitted to ICLR 2025_

### Official Review · Reviewer_BAgM · 2024-10-19

**Soundness:** 3
**Presentation:** 2
**Contribution:** 2
**Rating:** 5
**Confidence:** 5

**Summary:**

The paper aims to segment a wide range of concepts in video through a natural language. The paper proposes to rely on the powerful representation capabilities of the video diffusion model. Besides, to facilitate the research community, the paper introduces a new benchmark for Referral Video Process Segmentation (RVPS), which captures dynamic phenomena that exist at the intersection of video and language.

**Strengths:**

1. The focus on segmenting video processes presents a new research direction, which is valuable for exploring and evaluating model performance in this context.
2. The application of video diffusion to Referring Video Object Segmentation (RVOS) tasks is an interesting and promising approach.

**Weaknesses:**

1. The integration of video diffusion into referring segmentation appears incremental and lacks novelty, as it primarily adapts an existing video diffusion model to RVOS without specific tailoring for the task.
2. A detailed computational complexity analysis is needed to better understand the efficiency of the proposed method.
3. Both the figures and the overall writing quality require further refinement for better clarity and presentation.

**Questions:**

The definition of RVOS in the paper is **Referral** Video Object Segmentation in the introduction section, but it should be **Referring** Video Object Segmentation.

---

> ### Author Response · Authors · 2024-11-18
> **Response to Reviewer BAgM**
>
> Thank you for your review and valuable feedback. Please refer to our general response for a discussion of the computational requirements of our method (in short, the computational requirements of REM are on par with other approaches in the literature). We address the rest of your comments below.
>
> **W1: REM doesn’t tailor video diffusion models to the task of RVOS**
> Thank you for bringing up this important topic. Changing as little of the video diffusion model as possible is the core idea of our work and the reason behind the top performance in out-of-domain evaluation achieved by REM, not its limitation. This is discussed in detail in Section 3.2 and validated through ablation analysis in Section 5.3.
>
> Notably, VDIT [Zhu et al., ECCV’24] uses the same pre-trained video diffusion model as REM, but “tailors it to the task of RVOS”. It significantly underperforms compared to our method on all datasets precisely because of the modifications to the model architecture that it introduces.
>
> Finally, the simplicity of our method is highlighted as one of its main strengths by reviewers NxrS and bXbA.
>
> **W2: Presentation requires further improvement**
> We welcome concrete suggestions on how to refine the manuscript further and will incorporate any concrete feedback provided. However, we would like to note that reviewers NxrS, bXbA, and hMhL have commended the clarity and readability of our paper.
>
> **Q1: Referral vs Referring**
> We thank the reviewer for pointing out this discrepancy and have updated the manuscript accordingly.

---

> > ### Comment · Reviewer_BAgM · 2024-11-30
> > **Official Comment by Reviewer BAgM**
> >
> > Thanks for the author's response. While only some of my issues have been partially resolved, I have decided to increase my score to 5.

---

> > > ### Author Response · Authors · 2024-12-02
> > >
> > > Dear Reviewer BAgM,
> > >
> > > Thank you for revisiting your assessment and increasing your score. We appreciate your feedback. To ensure we fully address your remaining points, we kindly ask for more detail on the specific aspects where our response or revisions fell short.

---

> ### Author Response · Authors · 2024-11-22
> **Follow-up**
>
> Dear Reviewer BAgM,
>
> Thank you for your time and effort in reviewing our manuscript. We have carefully addressed all of your comments in our previous response. If our revisions and clarifications have resolved your concerns, we kindly ask that you acknowledge this and update your rating. Should you have any additional questions or require further clarification, we would be more than happy to address them. If additional experiments or results are needed, we kindly ask for sufficient time to ensure a thorough response.

---

### Official Review · Reviewer_hMhL · 2024-10-30

**Soundness:** 3
**Presentation:** 3
**Contribution:** 2
**Rating:** 5
**Confidence:** 4

**Summary:**

This paper explores using a pre-trained video diffusion model in referring segmentation. The paper proposes a new setting: Referral Video Process Segmentation, which is interesting. The model provides valuable insights on using generative prior knowledge to perform segmentation better.

**Strengths:**

1. The paper is well-written and easy to follow.

2. The task of Referral Video Process Segmentation is interesting.

3. The REM achieves comparable performance with the state-of-the-art method UNINEXT on the RES task and exhibits stronger performance on out-of-domain benchmarks.

**Weaknesses:**

REM uses a one-step approach to predict the segmentation mask during training. As a result, REM is more like resuming the pre-trained weights from the video diffusion model and training a U-Net segmentation model. I believe the model could achieve comparable performance without using the VAE Encoder to process the mask and instead using a few simple MLP layers. If this is the case, REM's performance gain may not be due to the authors' claim ``preserving as much of the generative model's original representation." Therefore, I think the authors need to add more detailed ablation studies to analyze the key factors leading to performance improvement.

**Questions:**

My biggest concern is listed in the Weaknesses.

---

> ### Author Response · Authors · 2024-11-18
> **Response to Reviewer hMhL**
>
> Thank you for your review and suggestions. An ablation analysis of our method is reported in Section 5.3 in the manuscript. This includes a variant of the ablation suggested here, where we replace the frozen VAE with a CNN mask decoder from [Zhao et al., ICCV’23] (second to last row in Table 4). This variant performs significantly worse than our default version, especially on the out-of-domain Ref-VPS benchmark. This result supports our claim that preserving the representation of a pre-trained video generation model is crucial for achieving strong generalization in referring video segmentation.
>
> Following your suggestion, we have evaluated a variant with an MLP mask decoder adopted from the popular SegFormer model [Xie et al., NeurIPS’21]. The results are reported below and added to Table 4 in the manuscript. Consistent with our findings for the CNN decoder, this variant also underperforms compared to the default configuration with a frozen VAE decoder.
>
> Decoder&nbsp; | &nbsp;Ref-YTB (J&F)&nbsp; | &nbsp;Ref-VPS (J)&nbsp;
> :---|:---:|:---:
> Ours | 64.57 | 37.80
> CNN | 60.47 | 25.09
> MLP | 59.35 | 31.75
>
> We hope this additional analysis addresses your concern. If you have any further suggestions for ablations, we would be happy to explore them.

---

> ### Author Response · Authors · 2024-11-22
> **Follow-up**
>
> Dear Reviewer hMhL,
>
> Thank you for your time and effort in reviewing our manuscript. We have carefully addressed all of your comments in our previous response. If our revisions and clarifications have resolved your concerns, we kindly ask that you acknowledge this and update your rating. Should you have any additional questions or require further clarification, we would be more than happy to address them. If additional experiments or results are needed, we kindly ask for sufficient time to ensure a thorough response.

---

> > ### Comment · Reviewer_hMhL · 2024-11-25
> >
> > Thanks for the author's response. The added ablation experiment have resolved my problem, and I decided to improve my score. However, due to the lack of sufficient analysis to provide some new insights, such as why using VAE to encode masks would bring improvement, I will only raise my score to 6.

---

> > > ### Author Response · Authors · 2024-11-26
> > >
> > > Dear Reviewer hMhL,
> > >
> > > We sincerely appreciate your thoughtful feedback and are glad the additional ablation experiments addressed your primary concern. Thank you for improving your score.
> > >
> > > We completely agree that understanding the inner workings of generative representations is both exciting and important. However, as you rightly pointed out, delving into the "how" and "why" questions requires extensive interpretability studies, which go beyond the scope of this paper. Our focus is on demonstrating the effectiveness of leveraging generative priors for referring video segmentation and establishing their strong generalization capabilities, as validated through extensive experiments.
> > >
> > > We do, however, offer an intuitive explanation of the effectiveness of our VAE-based approach. In particular, we hypothesize that encoding the masks into the latent space during training aligns well with the original video diffusion architecture, which was pre-trained to make predictions in the latent space of the VAE. This allows the U-Net to adapt more easily without significantly altering its pre-trained weights. In contrast, replacing the VAE with a new decoder (e.g., CNN or MLP) forces the U-Net to adapt its outputs to a new representation space, requiring more substantial changes to the pre-trained model. This misalignment explains the performance gap we observe, particularly on the out-of-domain Ref-VPS benchmark
> > >
> > > We hope this explanation helps to address your remaining concerns. Please let us know if there are any concrete experiments or analyses you would like to see, and we would be happy to explore them. Thank you again for your time and constructive feedback.

---

### Official Review · Reviewer_bXbA · 2024-11-02

**Soundness:** 3
**Presentation:** 3
**Contribution:** 2
**Rating:** 6
**Confidence:** 3

**Summary:**

The authors present a diffusion-model based method for the task of Referring Video Segmentation, called REM, and a new dataset for the task called Ref-VPS. The motivation for this work stems from the observation that current datasets for the task of Referring Video Segmentation are typically extensions of object-centric datasets for Referring Video Object Segmentation, which has prevented development of general video region segmentation methods. To address this, the authors create their carefully curated dataset, Ref-VPS, from TikTok videos, and label it with "concepts" that are not necessarily object-centric. They create REM from diffusion models that, due to their pretraining on Internet-scale data, have generalizable representations. The authors show that on Ref-VPS and two Referral Video Object Segmentation, REM either outperforms or performs favorably compared to current Referring Video Segmentation baselines. On an ablation study, the authors validate their claim that preserving the architecture of diffusion models during fine-tuning for Referring Video Segmentation leads to better performance.

**Strengths:**

1. REM has a straightforward design for training and inference that is presented clearly. In general the writing is clear.
2. REM results are strong on Ref-VPS and Ref-YTB and the ablation study effectively validates the author's claim that retaining the architecture of diffusion models is important for taking advantage of their generalizable representations for referring video segmentation. The finding that improving video generation models leads to enhanced video segmentation performance is interesting.

**Weaknesses:**

1. The motivation of the paper is not very clear. Specifically, it is not clear whether the authors are advocating for distinguishing the Referring Video Object Segmentation benchmarks from the Referring Video Process Segmentation benchmark that they propose. If the authors are advocating for this (and thus for two different tasks), then the final output of the model should be evaluated differently for each task, but this doesn't seem to be the case.
2. The Ref-VPS dataset was created from heavily curated TikTok videos and thus may not be representative of real-world scenes and videos. A more in-depth comparison of Ref-VPS to a dataset comprised of random videos (such as YouTube-BoundingBoxes [1]) would strengthen the paper if it is found that Ref-VPS adequately represents a wide variety of scenes, especially since the authors claim that the generalization ability of their method is a contribution.
3. The details of the REM method's computational efficiency and a comparison to that of the baselines are missing. Specifically, the training time, inference time and amount of memory use on the GPU should be included and compared to the baselines. The ease of use of the method is important in determining its contribution.

[1] "YouTube-BoundingBoxes: A Large High-Precision Human-Annotated Data Set for Object Detection in Video". Esteban Real, Jonathon Shlens, Stefano Mazzocchi, Xin Pan and Vincent Vanhoucke. CVPR, 2017.

**Questions:**

Questions:
1. Can you clarify whether you are proposing Referring Video Process Segmentation as a new/separate task?
2. What is the definition of "concept" and how is it different from a "class"? Is there high inter-annotator agreement on Ref-VPS? My concern is that the challenges of Referring Video Process Segmentation are due to the ambiguity of the task and the arbitrary nature of the label, rather than the lack of generalizable representations in the model.
3. In l. 157 can you clarify what you mean by "universal"? Do you mean generalizing to many domains?
4. In l. 184-188, what are the advantages of using such models? One can imagine that they are more lightweight or less computationally demanding than diffusion models.
5. In l. 275 what is t usually set to?
6. In l. 314 can you clarify what is meant by the "physical nature of the event"?
7. What architecture if REM specifically built on top of in the experiments in Section 5.1?
8. In Section 5.2 are the methods trained on the datasets as reported in Table 1?
9. In the "Fine-tuning strategy" section on page 10, out of curiosity, have you tried freezing the encoder and fine-tuning the decoder to preserve more of the representation?

Minor Typos/Writing Suggestions:
- l. 011 include what REM stands for
- l. 018 "crushing"->"crashing"
- "Referral" is used in l. 036, 039 but "Referring" is used in l. 134. It would be good to stay consistent to avoid confusion.
- l. 122 "approaches,"->"approaches"
- l. 160 "de-fact" -> "de-facto"
- l. 172 "highly-effective"-> "highly effective"
- l. 179 "RVOS,"->"RVOS"
- l. 260 "Figure 3 REM,"-> "Figure 3, REM"
- l. 263 "below"->"below."
- l. 313 "queries however" -> "queries; however"
- l. 331 "expression"->"expressions"
- l. 376 "video-segmentation"->"video segmentation"
- l. 377 "and we"->"and as we"
- l. 377 "by up to 46% out-of-domain in terms of Region Similarity" awkward wording
- l. 495 "StableDiffsuion"-> "StableDiffusion"

---

> ### Author Response · Authors · 2024-11-18
> **Response to Reviewer bXbA [1/3]**
>
> Thank you for your detailed review and suggestions. Please refer to our general response for a discussion of the computational requirements of our method during inference (in short, the computational requirements of REM are on par with other approaches in the literature). We address the rest of your comments below.
>
> **W1: Motivation of the paper**
> We are extending the task of Referring Video Segmentation (RVS) outside the domain of objects. As outlined in lines 36–44 of the paper, the RVOS community has focused on a limited subset of RVS, constrained to objects. Our work removes this artificial restriction, extending the task to include any spatio-temporally localizable entities in videos that can be described through natural language. Importantly, this extension does not alter the task itself but rather expands its applicability. Therefore, we believe the existing evaluation protocol remains adequate. However, we welcome any suggestions for additional evaluations that could enhance clarity or usefulness.
>
> **W2: Generalization to out-of-domain datasets other than Ref-VPS**
> Please note that we did evaluate REM on 2 existing out-of-domain datasets (BURST and VSPW) in the paper (see Table 2 and corresponding discussion in lines 405-419). Specifically, BURST is a large-scale video segmentation benchmark that focuses on object and scene diversity. It features 482 object categories, compared to 23 categories in YouTube-BoundingBoxes, and is labeled with segmentation masks. Our method outperforms the state-of-the-art methods on BURST by 9.5 points in terms of region similarity. In Figure E in the appendix we further investigate the performance of our method and the baselines on this dataset, demonstrating that our approach is indeed a lot more effective at capturing rare object categories in the tail of the distribution. These results provide strong evidence of REM's superior generalizability compared to prior approaches.
>
> **W3: Training costs**
> REM takes 174 hours to train on 4 A100 GPUs. Other methods do not report their training costs, so we have estimated them ourselves (excluding i/o time) given the same compute budget and report the results together with the memory consumption per GPU below, as well as in Table D in the manuscript:
>
> Method&nbsp; | &nbsp;MUTR&nbsp; | &nbsp;UNINEXT&nbsp; | &nbsp;VDIT&nbsp; | &nbsp;REM(ours)&nbsp;
> :---|:---:|:---:|:---:|:---:
> Memory (GB) | 30.4|30.2|68.5|61.8
> Runtime (hr) | 134|1096|260|174
>
> Our costs are on par with most prior works. Notably, UNINEXT – the state-of-the-art RVOS approach, takes 6.3 times longer to train than REM, since it utilizes more than 10 datasets with object supervision to achieve top object segmentation results. In contrast, REM effectively capitalizes on Internet-scale pre-training, allowing it to achieve competitive performance with UNINEXT on traditional RVOS dataset and significantly outperform it out-of-domain. All at a fraction of the training cost.

---

> > ### Comment · Reviewer_bXbA · 2024-12-02
> > **Generalization**
> >
> > Thanks for your detailed answers to the questions in my review, I appreciate it. My question and concern in W1 and W3 have been addressed.
> >
> > Regarding W2, BURST and VSPW are out-of-domain datasets to the TikTok videos in Ref-VPS but either there is enough overlap in objects and backgrounds of those datasets or BURST and VSPW domains are covered in the pretraining dataset, so that REM can perform well on those datasets. Would REM be able to perform well in a domain with no overlap, like a robotics or marine domain, by relying on the features learned and retained from the pretraining, or is it necessary to perform finetuning?

---

> > > ### Author Response · Authors · 2024-12-03
> > >
> > > Dear Reviewer bXba,
> > >
> > > Thank you for your thoughtful follow-up. We are glad that our response addressed most of your concerns and appreciate the opportunity to address the remaining issue below.
> > >
> > > * **Domain coverage in training vs. evaluation datasets**:
> > > We would like to clarify that our model is **only** trained on **94 object categories** from Ref-COCO and Ref-YTB. Datasets like Ref-VPS, BURST, and VSPW are used exclusively for evaluation. Therefore, the potential overlap between Ref-VPS and BURST or VSPW is not a factor in REM’s performance on these datasets. Notably, BURST contains 482 object categories, meaning that **388 of these categories are outside the training domain** of our model. The strong results on BURST, including a 9.5-point improvement in region similarity over state-of-the-art methods, highlight REM’s robust generalization to unseen objects.
> > > * **Role of generative pre-training**:
> > > We agree that determining what constitutes "out-of-domain" for Internet-scale datasets like LAION (used for generative pretraining) can be challenging. However, our results demonstrate that REM’s **superior generalization arises from our approach**, not solely from pretraining. Specifically, we strongly outperform VDIT—a method using the same generative representation—on all evaluation datasets, underscoring the effectiveness of our method in unlocking generalization.
> > > * **Performance in other domains**:
> > > We acknowledge your request to evaluate REM in robotics and marine domains. We have run REM on a few videos from these domains and added these results at the bottom of our [anonymous web page](https://refereverything.github.io/#ROBO). As you can see, the performance is **comparable to that in the domains shown in the paper**, suggesting that REM retains strong generalization without the need for domain-specific fine-tuning.
> > >
> > > We hope this addresses your remaining concern. If you have further suggestions or additional domains you'd like us to evaluate, we would be happy to explore those.
> > >
> > > Thank you again for your time and valuable insights!

---

> > > > ### Comment · Reviewer_bXbA · 2024-12-03
> > > > **Generalization**
> > > >
> > > > Thanks for clarifying that REM is only pre-trained on Ref-COCO and Ref-YTB and not pre-trained on large-scale internet data (correct me if I'm wrong). If this is the case then it addresses my concern in W2. I've increased my final score accordingly.
> > > >
> > > > That being said, I do think the amount of memory used by REM is quite cumbersome so I have some reservations, but since it is consistent with the current SOTA there is precedent.

---

> ### Author Response · Authors · 2024-11-18
> **Response to Reviewer bXbA [2/3]**
>
> **Q1: What is the definition of a concept and how do you handle ambiguity?**
> We use the term concept to refer to spatio-temporal entities that are semantically meaningful but cannot always be defined with a single word (i.e. they are more abstract than classes). The term goes back at least to [Ghorbani et al, NeurIPS’19; Kim et al., ICML’18], but, to the best of our knowledge, no formal definition still exists. We have added a reference to the best available definition in line 99 of the paper.
>
> Ambiguity is indeed an important aspect when annotating dynamic processes and we take it seriously. As mentioned in our response to reviewer NxrS above, and shown in Figure F in the manuscript, we have annotated all ambiguous regions with Ignore masks so that the evaluation is only performed on confident foreground and background pixels. Hence, the challenges of Ref-VPS are in fact due to generalizability, not due to the arbitrary nature of our annotations.
>
> **Q2: Meaning of “universal” in line 157**
> Yes, in this context we meant “universal” as not limited to one domain (e.g. not limited to common objects). We have clarified this in the manuscript to avoid confusion.
>
> **Q3: What are the advantages of visual representations learned with contrastive objectives?**
> These models indeed tend to be more lightweight (e.g. ViT used in CLIP vs the U-Net used by most diffusion models). We have updated the corresponding paragraph in Section 2 to mention this. Please note that, as discussed in the general response, the computational requirements of our model are on par with the state-of-the-art referring segmentation methods.
>
> **Q4: What is $t$ usually set to?**
> This depends heavily on the task and the architecture, but most works use low values of $t$ (between 0 and 100).
>
> **Q5: Meaning of “physical nature of the event” in line 314**
> For example, events like glaciers melting over time or soil erosion are not typically captured on TikTok due to the large temporal spans of such physical changes. We have added a clarifying text to the manuscript in lines 313-314.
>
> **Reference**
> [Ghorbani et al, NeurIPS’19] Towards Automatic Concept-based Explanations: https://arxiv.org/abs/1902.03129
> [Kim et al., ICML’18] Interpretability Beyond Feature Attribution: Quantitative Testing with Concept Activation Vectors (TCAV): https://arxiv.org/abs/1902.03129

---

> ### Author Response · Authors · 2024-11-18
> **Response to Reviewer bXbA [3/3]**
>
> **Q6: What architecture is REM built on top of?**
> We thank the reviewer for pointing out this missing detail. As described in our ablation in Section 5.3, we chose ModelScope T2V of Wang et al., as our base video diffusion architecture for all results in sections 5.1 and 5.2. Following the reviewer's comment, we have now explicitly mentioned this in the implementation details (lines 1116-1117).
>
> **Q7: Are the evaluations in Section 5.2 zero-shot?**
> As mentioned in lines 363-364 in the manuscript, all the evaluations in Section 5.2 are indeed 0-shot. Following the reviewer's comment, we have reiterated this at the beginning of Section 5.2.
>
> **Q8: Ablation with VAE decoder fine-tuning**
> We thank the reviewer for suggesting this insightful experiment. The results are reported below. The proposed variant with a fine-tuned VAE decoder performs worse than our proposed approach with a frozen VAE decoder, but better than a CNN decoder trained from scratch on our Ref-VPS benchmark. This result further supports our claim that preserving the representation of a pre-trained video-diffusion model is key for achieving strong generalization in referring video segmentation.
>
> Decoder | &nbsp;Ref-YTB (J&F)&nbsp; | &nbsp;Ref-VPS (J)&nbsp;
> :---|:---:|:---:
> Ours | 64.57 | 37.80
> CNN | 60.47 | 25.09
> Fine-tuned VAE | 55.08 | 30.12
>
> **Q9: Typos and writing suggestions**
> We greatly appreciate the reviewer’s attention to detail and have implemented all suggested corrections to improve the manuscript’s readability.

---

> ### Author Response · Authors · 2024-11-22
> **Follow-up**
>
> Dear Reviewer bXbA,
>
> Thank you for your time and effort in reviewing our manuscript. We have carefully addressed all of your comments in our previous response. If our revisions and clarifications have resolved your concerns, we kindly ask that you acknowledge this and update your rating. Should you have any additional questions or require further clarification, we would be more than happy to address them. If additional experiments or results are needed, we kindly ask for sufficient time to ensure a thorough response.

---

> > ### Author Response · Authors · 2024-12-02
> >
> > Dear Reviewer bXbA,
> >
> > We hope this message finds you well. We wanted to kindly remind you that the reviewer response deadline is at midnight today. We would greatly appreciate it if you could let us know whether your concerns have been addressed or if there are any additional questions or suggestions we can assist with.
> >
> > To further support your evaluation, we have conducted an ablation study on the impact of the time step $t$ on the performance of our model for Ref-YTB and Ref-VPS. The results, detailed below, demonstrate that while our choice of $t=0$ is indeed optimal, the impact of this hyperparameter on performance is relatively minor. We will include this ablation in the camera-ready version of the manuscript.
> >
> > Noise level (t) | Ref-YTB (J&F) | Ref-VPS (J)
> > ---|---|---
> > t=200 | 59.2 | 36.2
> > t=50 | 62.9 | 35.0
> > Ours (t=0) | 64.6 | 37.8

---

> ### Author Response · Authors · 2024-12-03
>
> Dear Reviewer bXba,
>
> Thank you for the constructive discussion and for updating the score. We sincerely appreciate it.
>
> To clarify, while REM is trained **only** on segmentation data from Ref-COCO and Ref-YTB, it leverages a video diffusion backbone trained on Internet-scale data with a self-supervised, generative objective. This approach is consistent with methods like VDIT, which uses the same backbone. However, our method demonstrates significantly stronger generalization compared to VDIT, as shown in our experiments on Ref-VPS, BURST, and VSPW (relative improvement of up to 28%). These results highlight the effectiveness of our approach in unlocking the generalization potential of pre-trained video diffusion models.

---

### Official Review · Reviewer_NxrS · 2024-11-02

**Soundness:** 3
**Presentation:** 3
**Contribution:** 2
**Rating:** 6
**Confidence:** 4

**Summary:**

This paper presents a method to adapt a video diffusion model (a text-to-video generative model) into a referral video segmenter (a text+video to masklet model).
Instead of segmenting clearly defined objects as conventionally done in image and video object segmentation, the authors focus on segmenting dynamic concepts such as a wave or a cloud of smoke. In order to evaluate their model on this task, the authors introduce the Referral Video Process Segmentation (Ref-VPS) benchmark, consisting of 111 annotated Tiktok videos.
Because of the lack of training data for segmenting such dynamic "entities", the authors propose to use a video diffusion model, which has learned the mapping from linguistic descriptions to video on massive amounts of text-video pairs. The model is very minimally adapted to the Referral Video Segmentation task in order to conserve its capabilities acquired during pretraining.

**Strengths:**

The paper tackles an interesting problem in video segmentation.
The approach is straightforward and well-motivated, leveraging existing diffusion models.
The contributions are clearly articulated, and the paper is well-structured.
The authors commit to releasing code, models, and data, which supports reproducibility of their results by the community.

**Weaknesses:**

Evaluation of Ref-VPS:
Dynamic concepts such as a wave or a cloud of smoke don't have clearly defined edges. Providing a single ground-truth contour for them is necessarily arbitrary. How then to evaluate?
For example, in Figure 1, the mask for “the smoke dissipating” arguably does not cover all the smoke. Similarly, for ”the wave crashing in the ocean”, the ground-mask does not include the wave foam on the left. A model prediction that includes these elements would be unjustly penalised.
Segmentations on this kind of "dynamic concepts" are notoriously hard to evaluate but this issue is mostly not covered in this paper. The authors exclude the contour accuracy F metric but the J metric also suffers from this issue. There are possible solutions to alleviate this problem and their discussion is expected for a paper introducing a benchmark on segmenting "Dynamic concepts".

Runtime:
In order to be practical, the approach presented in this paper should have a reasonable running time. The paper does not give details on this subject.
For example, how long does it take for the model to segment a video of 256 frames, using an A100 GPU?

**Questions:**

In order to match the pre-training input distribution of the model, the authors add noise to the input (as performed during pre-training).
However, it corrupts the input image and reduces the information amount accessed by the model.
Have the authors experimented with not adding noise to the input? That would simplify the approach further.

How many frames does the model process at once? (What is the value of T ?)

Minor typos:
L309: "enities"
L386: "UINNEXT"
L495: "StableDiffsuion"
L523: "changing a little as possible"

---

> ### Author Response · Authors · 2024-11-18
> **Response to Reviewer NxrS**
>
> Thank you for your detailed review and suggestions. Please refer to our general response for a discussion of the computational requirements of our method (in short, the computational requirements of REM are on par with other approaches in the literature). We address the rest of your comments below.
>
> **W1: Boundaries of dynamic concepts**
> Firstly, we would like to clarify that Figure 1 shows REM predictions, **not ground-truth**. We have updated the caption of the figure to make this clear. Secondly, we fully acknowledge that segmenting dynamic concepts like smoke or waves introduces inherent challenges due to ambiguous boundaries. To address this, Ref-VPS incorporates Ignore masks in the annotations, which mark regions of uncertainty that are excluded during quantitative evaluations. This approach ensures that the metrics focus on evaluating the most reliable regions of the masks, avoiding arbitrary penalties for ambiguous boundaries. We have added examples of our ground-truth annotations, including Ignore masks in Figure B in the manuscript together with a corresponding discussion. We appreciate the reviewer highlighting this critical aspect, and we have ensured that the manuscript includes detailed descriptions and examples of our evaluation strategy.
>
> **Q1: Noise level for the diffusion model**
> This is a great point! To preserve input fidelity while leveraging the benefits of pre-training, we do in fact set the noise level to 0, as mentioned in lines 272–275 of the manuscript. This ensures that the input images retain as much information as possible without being corrupted. Following the reviewer's suggestion, we have further highlighted this in the implementation details section (line 1117).
>
> **Q2: How many frames does the model process at once?**
> Thank you for pointing out this missing detail. We set the number of frames to 72 at inference time, a value determined by GPU memory constraints. We have updated the implementation details section in the manuscript accordingly (lines 1124-1125).
>
> **Q3: Typos**
> We thank the reviewer for the detailed feedback and have updated the manuscript accordingly.

---

> ### Author Response · Authors · 2024-11-22
> **Follow-up**
>
> Dear Reviewer NxrS,
>
> Thank you for your time and effort in reviewing our manuscript. We have carefully addressed all of your comments in our previous response. If our revisions and clarifications have resolved your concerns, we kindly ask that you acknowledge this and update your rating. Should you have any additional questions or require further clarification, we would be more than happy to address them. If additional experiments or results are needed, we kindly ask for sufficient time to ensure a thorough response.

---

> > ### Author Response · Authors · 2024-12-02
> >
> > Dear Reviewer NxrS,
> >
> > We hope this message finds you well. We wanted to kindly remind you that the reviewer response deadline is at midnight today. We would greatly appreciate it if you could let us know whether your concerns have been addressed or if there are any additional questions or suggestions we can assist with.
> >
> > To further support your evaluation, we have conducted an ablation study on the impact of the time step $t$ on the performance of our model for Ref-YTB and Ref-VPS. The results, detailed below, demonstrate that while our choice of $t=0$ is indeed optimal, the impact of this hyperparameter on performance is relatively minor. We will include this ablation in the camera-ready version of the manuscript.
> >
> > Noise level (t) | Ref-YTB (J&F) | Ref-VPS (J)
> > ---|---|---
> > t=200 | 59.2 | 36.2
> > t=50 | 62.9 | 35.0
> > Ours (t=0) | 64.6 | 37.8

---

### Official Review · Reviewer_QTA2 · 2024-11-05

**Soundness:** 3
**Presentation:** 3
**Contribution:** 3
**Rating:** 6
**Confidence:** 3

**Summary:**

The REM paper introduces a novel framework for Referral Video Segmentation (RVS) that uses a video-text diffusion model to generalize segmentation to a wide range of visual concepts.REM effectively segments both common and rare objects, as well as non-object dynamic concepts like smoke, water, or changing light. The paper also introduces Ref-VPS, a new segmentation benchmark on dynamic processes that require both spatial and temporal modeling.

**Strengths:**

* REM achieves good generalization to out-of-domain concepts and non-object entities by leveraging Internet-scale pretraining on video-language data, outperforming specialized methods.

* The Ref-VPS benchmark provides a way to evaluate models on dynamic processes, filling a gap in existing RVS datasets by focusing on temporal events and continuous appearance changes.

**Weaknesses:**

* The REM method performs comparably to MUTR  and only slightly better then VLMO on Ref-YTB.

* Given use of a high-capacity diffusion model, there is limited discussion on its computational efficiency or inference speed. A comparison of REM's computational requirements with prior methods would be beneficial.

* Analysis on the impact of the quantity and quality of synthetic data would be useful - whether reducing/increasing the amount of synthetic data affect REM's performance.

* Since REM is applied to video segmentation tasks, especially focusing on continuous changes, it would be great to have some evaluation of temporal consistency.

**Questions:**

* How does REM handle ambiguous or overlapping concepts (e.g., scenes where multiple dynamic elements, like water and smoke, interact)?

* How critical is the quality of synthetic data for REM’s generalization across diverse concepts?

---

> ### Author Response · Authors · 2024-11-18
> **Response to Reviewer QTA2 [1/2]**
>
> Thank you for your detailed review and suggestions. Please refer to our general response for a discussion of the computational requirements of our method (in short, the computational requirements of REM are on par with other approaches in the literature). We address the rest of your comments below.
>
> **W1: Performance on Ref-YTB**
> We would like to emphasize that our model is not specifically designed for video segmentation and instead re-uses a video generation model’s architecture in its entirety. Despite this, it performs on par with specialized approaches like MUTR on datasets for which they were designed and heavily tuned. In particular, it clearly outperforms MUTR in terms of region similarity on both Ref-DAVIS and Ref-YTB. More importantly, it dramatically outperforms these approaches out of the domain (relative improvement of up to 51% with respect to MUTR), both on object-centric datasets like BURST and on non-object centric ones like Ref-VPS.
>
> VLMO-L also capitalizes on Internet-scale pre-training, but our approach is more effective at doing so on RVOS datasets. Unfortunately, neither pre-trained models, nor training details are available for this (unpublished) method, so we cannot compare to it out-of-domain. That said, based on the fact that VLMO-L shares a similar architecture with VDIT, with the main difference between the two methods being that the former adopts a VLM backbone, whereas the latter is based on a video diffusion one, we can expect that VLMO-L would also struggle to generalize.
>
> To sum up, our method offers unique out-of-domain generalization, while being highly competitive with specialized approaches in the narrow domain of traditional RVOS benchmarks. Moreover, the performance of REM is likely to further improve with the advancement of video generation models, as suggested by our analysis in Section 5.3.
>
> **W2: Impact of synthetic data**
> We would like to clarify that no synthetic data was used in training REM. Instead, as described in lines 355–357 of the manuscript, we follow prior work and train our model on real-world Ref-COCO and Ref-YTB datasets only. Please let us know if you have further questions regarding our training protocol.
>
> **W3: Temporal consistency evaluation**
> We thank the reviewer for this insightful suggestion. Unfortunately, accurately evaluating the temporal consistency of video segmentation methods is notoriously challenging, because it is hard to distinguish between predicted mask changes that are due to the method’s inconsistency and the changes that are due to the true target deformations. Notably, the temporal consistency metric proposed in the original DAVIS dataset was only applied to videos with no significant object deformations and no occlusions and was eventually phased out by the dataset’s authors.
>
> Recognizing these limitations, we implemented a straightforward consistency metric by computing the average difference of IoU between the model's prediction and the ground truth mask in consecutive frames. Formally,
>
> $$
> \text{Temp.~Con.} = \frac{1}{N}  \sum_{n=1}^N \left [\frac{1}{T_n} \sum_{t=1}^{T_n}(IoU(Pred_{t+1}, GT_{t+1}) - IoU (Pred_t, GT_t) \right ],
> $$
>
> where $N$ is the number of samples and $ T_n$ is the number of frames in the n-th sample.
> **Lower numbers indicate better temporal consistency on this metric**, and it’s easy to see that simply outputting empty masks would result in the perfect consistency score of 0. Hence, as with any temporal consistency metric, it should always be considered jointly with a prediction accuracy metric.
>
> We report region similarity and temporal consistency on Ref-VPS and Ref-DAVIS (the two datasets that extract frames at 24 fps) below, as well as in Table B in the updated manuscript. The results demonstrate the superior temporal consistency of REM on both object-centric and non-object-centric datasets. Notably, UNINEXT - the state-of-the-art RVOS approach, shows the worst temporal consistency out of all methods. MUTR achieves a strong temporal consistency score on Ref-VPS precisely because it often outputs empty predictions, as can be seen from its low region similarity score.
>
>  Method| &nbsp;J (Ref-VPS)&nbsp; | &nbsp;Temp. Con. (Ref-VPS)&nbsp; | &nbsp;J (Ref-DAVIS)&nbsp; | &nbsp;Temp. Con. (Ref-DAVIS)&nbsp;
>  :---|:---:|:---:|:---:|:---:
>  MUTR | 24.1 | 2.9 | 64.8 | 3.4
> UNINEXT | 26.3 | 5.2 | 68.2 | 5.2
> VDIT | 35.3 | 4.7 | 66.2 | 3.1
> REM(ours) | 49.0 | 2.8 | 69.9 | 2.1

---

> ### Author Response · Authors · 2024-11-18
> **Response to Reviewer QTA2 [2/2]**
>
> **Q1: How does REM handle ambiguous or overlapping concepts?**
> This is a great question! Our Ref-VPS indeed contains some ambiguous scenarios and we took great care in labeling them accurately (see our response to Reviewer NxrS below as well as the new Figure B in the manuscript). We have added a few examples of REM outputs on ambiguous or overlapping concepts in Figure F in the appendix and compared our method with VDIT - the strongest baseline on this benchmark. As you can see, although for many of these samples, no perfect prediction exists, the outputs of our model are both more accurate in the confident regions and more consistent. For example, in the first row, our method only segments the clearly visible regions of lava once it is hit by a wave, whereas VDIT segments the entire wave as well. In the second row, REM consistently segments all the glowing water, whereas VDIT only covers a few patches.

---

> ### Author Response · Authors · 2024-11-22
> **Follow-up**
>
> Dear Reviewer QTA2,
>
> Thank you for your time and effort in reviewing our manuscript. We have carefully addressed all of your comments in our previous response. If our revisions and clarifications have resolved your concerns, we kindly ask that you acknowledge this and update your rating. Should you have any additional questions or require further clarification, we would be more than happy to address them. If additional experiments or results are needed, we kindly ask for sufficient time to ensure a thorough response.

---

### Author Response · Authors · 2024-11-18
**General response**

We thank the reviewers for their feedback and valuable suggestions, which helped us further strengthen the paper. We are glad that reviewers found our approach to be **straightforward and well-motivated** (reviewers NxrS and bXbA), its **zero-shot out-of-domain generalization** performance to be strong (reviewers QTA2, bXbA, and hMhL), the proposed extension of **referring video segmentation outside of the domain of objects** to be valuable (reviewers QTA2, NxrS, hMhL, and BAgM) and the paper to be **well written** (reviewers NxrS, bXbA, and hMhL). We have addressed all the comments individually, and highlight a couple of common concerns in this general response.


### **Ground-truth annotations**
We would like to emphasize that Figure 1 in the manuscript shows REM predictions, not the ground-truth. We have clarified this in the caption to avoid confusion. While our model’s predictions are accurate, the ground-truth annotations in the Ref-VPS benchmark offer even greater precision, and explicitly address the natural ambiguity of the problem with Ignore labels. We have added Figure B with a corresponding discussion to the appendix to illustrate this. In particular, the entire extent of the wave is labeled in the third row. Rows 1 and 4 illustrate the handling of ambiguous scenarios, where only the confident regions of the glowing water and of the light column are labeled as target, and the ambiguous regions are labeled as Ignore (shown in gray). Pixels inside the Ignore regions are not included in the metric calculation.

### **Inference costs**
Several reviewers have raised concerns about the computational complexity of our approach. While REM is based on a video diffusion model, which is inherently computationally intensive due to its multiple inference steps, we emphasize that REM is trained and evaluated as a feed-forward model for referring video segmentation. Consequently, its memory requirements and runtime are comparable to those of other methods in the literature.

We report the inference speed and the memory consumption of REM alongside the primary baselines from our paper using their public implementations below:

Model | Inference speed (FPS)  | Memory (GB)
:---|:---:|:---:
MUTR| 13.6 |34.1
UNINEXT| 3.3 |9.7
VDIT| 7.1 |72.8
Ours| 7.1 |41.8

These values are estimated under the following protocol: inference was performed on 32-frame clips from Ref-DAVIS on a single A100 GPU with averages computed over 80 runs. As shown, the inference costs of REM align with those of other state-of-the-art approaches. We have incorporated these results into Table C in the manuscript.

---

### Author Response · Authors · 2024-12-04
**Final response**

Dear Reviewers,

We deeply appreciate your thoughtful feedback and constructive engagement, which helped to improve our paper. Below, we summarize the key contributions of our work and highlight the additional analyses provided in the rebuttal, which address your comments and demonstrate the strengths of our approach:


**Novel Contributions:**
- We introduce **REM**, a Referring Video Segmentation (RVS) framework that can segment a wide range of object-centric and non-object-centric concepts in video that can be described through natural language, while being trained **only on standard referring object segmentation datasets** (Ref-COCO and Ref-YTB).
- By capitalizing on visual-language representations learned by video diffusion models, REM achieves strong generalization to **out-of-domain datasets** (BURST, VSPW, and Ref-VPS), outperforming both traditional approaches, and methods like VDIT that share the same backbone, by up to 28% in terms of relative improvement.
- We present **Ref-VPS**, the first benchmark dedicated to **dynamic processes** involving spatiotemporal changes beyond object-centric segmentation tasks. Ref-VPS evaluates **zero-shot** model generalization to challenging dynamic concepts, filling a critical gap in the existing literature.


**Key Insights from Rebuttal Results:**
- **Generalization to Unseen Domains:** The superior performance of REM across BURST’s 482 diverse object categories, despite only training on 94 categories, underscores the method’s ability to generalize beyond the training domain.
- **Inference Speed and Compute:** REM’s inference speed and compute are in line with the state-of-the-art methods, while delivering much better OOD generalization.
- **Impact of Hyperparameters:** Ablations on the mask decoder design and time-step parameter $t$ confirmed that our choices are optimal.
- **Temporal Consistency:** We implemented and reported a new temporal consistency metric in the rebuttal, showing REM’s superior ability to model continuous changes compared to state-of-the-art baselines.
- **Clarifications on Ref-VPS:** We take great care to ensure robust evaluation of challenging dynamic concepts in Ref-VPS by labeling all ambiguous regions as Ignore.
- **Clarity of the presentation:** We have updated the manuscript to address all specific suggestions regarding clarity and writing.


**Conclusion:**
Our work pioneers a novel direction in RVS, with REM delivering unparalleled generalization across diverse and dynamic scenarios while introducing Ref-VPS as a new, valuable benchmark. We hope the provided results and clarifications demonstrate the rigor, impact, and potential of this research.

---

### Meta-Review · Area_Chair_8zcU · 2024-12-16

**Metareview:**

This paper receives mixed reviews from five reviewers.

The positive reviewers think the new setting is interesting, and it evaluates models on dynamic processes.

The negative reviewers were concerned about the novelty of technical parts, ablation study details, and several writing problems in particular for the main highlighted contributions.

During the rebuttal, the authors have give detailed feedbacks on several aspects, including generalization to unseen domains (more experiments), inference time, temporal consistency, more ablation studies on hyper-parameters and improved writing.  Parts of these issues are solved. One reviewer raises the score from 3 to 5. One reviewer raises the score from 4 to 6.  However, after discussion with the reviewers, there are still two negative reviews in the final stage.

The AC have checked the refined version of the submission and agrees with the negative sides.

The next submission must include a more detailed comparison with existing diffusion-based segmentation methods, more clear motivation, a clearer contribution statement, more detailed ablation studies, and future works on bad cases.

**Additional Comments On Reviewer Discussion:**

None

---

### Decision · Program_Chairs · 2025-01-22

Reject